# Homeostatic serum IgE is secreted by plasma cells in the thymus and enhances mast cell survival

Dong-il Kwon[1], Eun Seo Park[2], Mingyu Kim[1], Yoon Ha Choi[1,2], Myeong-seok Lee[3], Si-hyung Joo[3], Yeon-Woo Kang[1], Minji Lee[1], Saet-byeol Jo[1], Seung-Woo Lee[1], Jong Kyoung Kim [1,2✉] & You Jeong Lee [1,3✉]

Increased serum levels of immunoglobulin E (IgE) is a risk factor for various diseases, including allergy and anaphylaxis. However, the source and ontogeny of B cells producing IgE under steady state conditions are not well defined. Here, we show plasma cells that develop in the thymus and potently secrete IgE and other immunoglobulins, including IgM, IgA, and IgG. The development of these IgE-secreting plasma cells are induced by IL-4 produced by invariant Natural Killer T cells, independent of CD1d-mediated interaction. Single-cell transcriptomics suggest the developmental landscape of thymic B cells, and the thymus supports development of transitional, mature, and memory B cells in addition to plasma cells. Furthermore, thymic plasma cells produce polyclonal antibodies without somatic hypermutation, indicating they develop via the extra-follicular pathway. Physiologically, thymic-derived IgEs increase the number of mast cells in the gut and skin, which correlates with the severity of anaphylaxis. Collectively, we define the ontogeny of thymic plasma cells and show that steady state thymus-derived IgEs regulate mast cell homeostasis, opening up new avenues for studying the genetic causes of allergic disorders.

[1] Department of Life Sciences, Pohang University of Science and Technology (POSTECH), Pohang 37673, Republic of Korea. [2] Department of New Biology, DGIST, Daegu 42988, Republic of Korea. [3] Research Institute of Pharmaceutical Sciences, College of Pharmacy, Seoul National University, Seoul 08826, Republic of Korea. ✉email: blkimjk@postech.ac.kr; youjeonglee@snu.ac.kr

Immunoglobulin E (IgE) antibodies are the main mediators of allergic disorders[1,2], but also provide protection against helminth infections in both mice and humans[3]. They also play a role in host defense against venoms[4], and xenobiotic-mediated cancer[5] in mice. Serum IgE concentrations are tightly regulated in healthy individuals and elevated in various disease conditions, including autoimmunity[6], immuno-deficiency[7], and perturbed commensal microbiota[8]. However, the regulatory mechanisms of homeostatic serum IgE levels are largely unknown. A previous study showed that IgE$^+$ B cells develop from IgG1$^+$ germinal center (GC) intermediates and are excluded from follicles[9]. In contrast, other studies using IgE reporter mice showed that B cells were rapidly class-switched into IgE isotype in the GCs and prone to differentiate into short-lived plasma cells (SLPCs)[10,11]. However, these studies used mice immunized with exogenous antigens, and the development of IgE-secreting B cells by endogenous self-antigens has not been addressed.

Under steady state conditions, BALB/c mice displayed 100-fold higher serum IgE levels together with a higher frequency of IL-4-producing NKT2 cells than B6 mice in the thymus[12–14]. However, the source of homeostatic IgE-producing B cells in BALB/c mice has been enigmatic. As IgE class-switching requires IL-4 and the frequency of NKT2 cells is highest in the thymus[13], we hypothesized that they regulate the development of IgE-producing plasma cells (PCs). Thymus harbors tissue-resident B220$^+$CD19$^+$ B cells[15], and their cognate interaction with CD4 T cells induces licensing and class switch recombination (CSR), which deletes autoreactive CD4 T cells by expressing AIRE[16–18]. Nevertheless, little is known about the differentiation of these B cells into antibody-secreting PCs.

Here, we show that CD138$^+$Blimp1$^+$ PCs normally reside in the thymus and contribute to basal serum levels of IgM, IgG, and IgA. These cells originate from post-natal BM cells as early as 1 week after birth independently of exogenous antigens. In the presence of copious amounts of IL-4 from NKT2 cells, IgE$^+$ PCs also developed. Single-cell RNA sequencing (scRNA-seq) paired with B cell receptor (BCR) repertoire analysis revealed multistages of thymic B cell development from transitional B cells to PCs expressing AIRE without somatic hypermutations (SHMs). Finally, we show that thymus-derived IgEs increase the numbers of mast cells (MCs) in the gut and skin, enhancing allergic immune responses. Collectively, we define a previously unrecognized immune axis between thymus and peripheral tissues mediated by IgE.

## Results

**IgE-producing PCs develop in the thymus.** Compared to B6 mice, BALB/c mice have a prominent population of PLZF$^{hi}$ NKT2 cells in the thymus, which secrete homeostatic IL-4 and condition CD8 T cells to be memory-like[12]. We confirmed this again by staining IL-4 in WT mice or human CD2 (hCD2) in KN2 mice, which knocked-in hCD2 at the endogenous IL-4 locus, thereby expressing hCD2 on the cell surface when cells produce IL-4[19] (Supplementary Fig. 1A-C). We further validate that IL-4 expression was mainly detected in PLZF$^{hi}$Tbet$^-$RORγt$^-$ NKT2 cells using KN2 mice crossed with Tbet-Zsgreen mice, which are bacterial artificial chromosome (BAC) transgenic T-bet reporter mouse[20] (Supplementary Fig. 1D). Consistent with this, BALB/c mice have significantly higher serum levels of Th2 type immunoglobulins (Igs) (IgE and IgG$_1$) and lower Th1 type Igs (IgG$_{2b}$ and IgG3) compared to B6 mice (Supplementary Fig. 2A). Age-dependent kinetics of serum IgE levels in BALB/c mice showed that serum IgE is detected as early as 1 week after birth and gradually increases with age (Fig. 1A). In $Cd1d^{-/-}$ BALB/c mice, there was a specific decrease of serum IgE levels about 5-folds

compared to WT control (Fig. 1A), without affecting other isotypes (Supplementary Fig. 2B). To track the localization of IgE secreting B cells in BALB/c mice, we analyzed mature secreted and switched intermediate forms of Cε transcripts[9] in various lymphoid organs. Interestingly, they were most frequently detected in the thymus of BALB/c mice (Supplementary Fig. 2C), and, along with this, serum IgE levels decreased approximately 4-fold 3 weeks after thymectomy (Fig. 1B). Notably, not only IgE, but also serum concentrations of IgM, IgA, IgG$_1$, and IgG$_{2b}$ were decreased in thymectomized mice (Supplementary Fig. 2D). Furthermore, depletion of peripheral CD4 T cells did not affect serum IgE concentrations (Supplementary Fig. 2E-F), suggesting that thymus contains B cells that provide homeostatic Igs.

Previous reports showed that class-switched B cells are found in the thymus[16,18]. However, IgE$^+$ B cells have not been identified. Thus, we analyzed PCs by enriching them using anti-CD138 microbeads and performed IgE intracellular staining that has been shown to have the same sensitivity and specificity as those of fluorescence IgE reporter mice[11]. We also validated that anti-IgE antibodies do not have cross-reactivity to other isotypes (Supplementary Fig. 3A, B). Surprisingly, we found CD138$^+$Blimp1$^+$ PCs are present in the thymus, and they had either surface IgM (sIgM) or intracellular (ic) IgA, IgG, or IgE (Fig. 1C and Supplementary Fig. 3C). Their mean fluorescent intensities (MFIs) of icIgs were substantially higher than those of CD138$^-$B220$^+$ thymic B cells (Supplementary Fig. 3D), and their antibody-producing capacities were comparable to those of OVA-alum immunized splenic PCs (Supplementary Fig. 4A–C). Consistent with qPCR analysis (Supplementary Fig. 2C), the numbers of icIgE$^+$ thymic PCs were dramatically decreased in $Cd1d^{-/-}$ and $Il4^{-/-}$ ($Il4^{KN2/KN2}$) BALB/c mice compared to WT control (Fig. 1C), whereas the numbers of IgA$^+$ and IgG$^+$ thymic PCs were unaffected (Fig. 1D). Furthermore, we found that IgE$^+$ PCs were almost completely disappeared in $Il4ra^{-/-}$ BALB/c mice, indicating that differentiation of IgE$^+$ PCs in the thymus also partly relys on IL-13 signaling (Fig. 1C, D). icIgE$^+$ PCs were rarely detected outside the thymus (Fig. 1E and Supplementary Fig. 5), and we further confirmed that sorted CD138$^+$ PCs, but not CD138$^-$B220$^+$ B cells, express Cε transcripts (Fig. 1F) and produce IgE in in vitro culture (Fig. 1G). Compared to splenic PCs, thymic PCs highly expressed markers associated with activation such as MHC II and CD80 and PC differentiation such as CD138, CD93, Ly6C, EpCAM, CXCR3, CXCR4, and IRF4[21] (Fig. 1H). Taken together, these results suggest that the development of IgE-producing PCs depends on IL-4 from iNKT cells in the thymus.

**iNKT cells regulate the development of IgE$^+$ thymic PCs.** The number of icIgE$^+$ thymic PCs was correlated with serum IgE concentrations and NKT2 cell counts in mice between 1 and 7 weeks of age (Fig. 2A). Because the number of NKT2 cells in BALB/c mice increases as the mice ages[12], this correlation could be a result of aging. To confirm this correlation in an age-independent manner, we generated mice that had a variable fraction of NKT2 cells at the same age groups; F1 mice between BALB/c and B6 mice, which have high and low frequencies of NKT2 cells, respectively, were bred with their parental mice (Fig. 2B). F1 mice have the same phenotype as BALB/c mice for their iNKT cell profiles, whereas the frequency of NKT2 cells in B6 X F1 mice averaged between the median values of B6 and BALB/c mice. In these mice, the frequencies of IgE$^+$ thymic PCs correlated with those of NKT2 cells, while IgA$^+$ or IgG$^+$ PCs did not (Fig. 2C). Although indirectly tested, these results suggest that IL-4 produced from NKT2 cells regulates the development of IgE-producing PCs in the thymus.

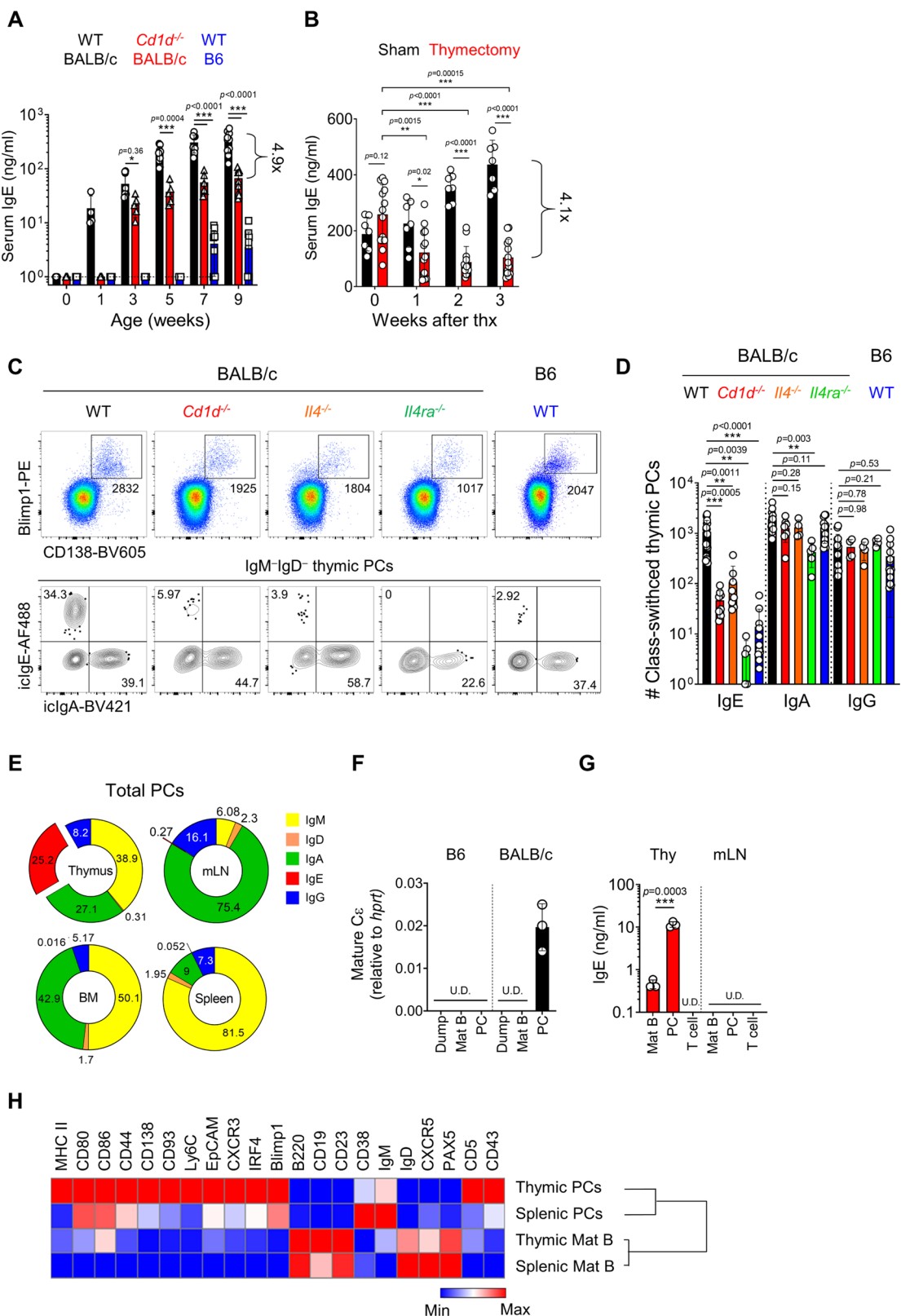

To address whether a CD1d-mediated interaction between iNKT cells and B cells is required for the development of IgE⁺ PCs, we generated mixed BM chimeras as depicted in Fig. 2D. In WT + $Cd1d^{-/-}$→ WT chimeras, the absence of CD1d molecules on the surface of B cells did not affect the development of thymic PCs or their IgE CSR. However, in WT + $Il4ra^{-/-}$→ WT chimeras, IgE⁺ thymic PCs did not develop at all from IL-4Rα-

deficient donor cells, while PCs with other isotypes were unaffected (Fig. 2E). Overall, these results indicate that IL-4, but not CD1d-mediated interactions, is essential for the differentiation of IgE-producing thymic PCs.

To further validate whether NKT2 cells directly induce the development of intrathymic IgE⁺ PCs, we performed fetal thymus organ culture (FTOC) experiments as a closed system

**Fig. 1 IgE-producing PCs develop in the thymus. A** Graph shows serum IgE levels in B6 and WT and $Cd1d^{-/-}$ BALB/c mice measured by ELISA at the indicated ages ($N = 6$-9). Dotted lines indicate the detection limit. **B** Graph shows serum IgE concentrations in sham-operated and thymectomized BALB/c mice ($N = 13$) measured by ELISA. **C, D** Total thymocytes of B6 and BALB/c mice were enriched for CD138$^+$ cells by MACS and stained with indicated markers. Representative dot plots are shown after gating out TCRβ$^+$ cells (**C**). Numbers indicate the total number of cells in adjacent gates (upper) or the frequency of cells in each quadrant (lower). Graph shows numbers of icIgE, icIgA and icIgG$^+$ PCs and statistical analysis ($N = 4$-19) (**D**). Results are from more than three independent experiments. **E** Pie charts show mean frequencies of each isotype of PCs in the thymus, mLN, BM, and spleen ($N = 3$-4). Numbers indicate mean values of frequencies. **F, G** Indicated cells were purified by MACS enrichment followed by FACS sorting. CD3$^+$CD4$^+$CD11b$^+$ cells were used as dump cells. Graph shows relative levels of mature Cε transcripts normalized to *hprt* by qPCR ($N = 3$) (**F**). Indicated cells were cultured for 5 days, and total IgE concentrations were measured in the supernatant ($N = 3$) (**G**). **H** Heat map shows MFIs of indicated markers in B220$^+$CD19$^+$ mature B and PCs in thymus and spleen measured by flow cytometry. Data were log2 transformed and visualized by relative expression per column. Data are presented as mean values ± SD (**A, B, D, F,** and **G**). Each dot represents an individual mouse. Unpaired two-tailed *t*-test (**A, B,** and **G**) and one-way ANOVA (**D**) were used for comparison. $^{***}p < 0.001$, $^{**}p < 0.01$, $^*p < 0.05$. Not significant ($p > 0.05$). Thx Thymectomy; U.D. Undetected; PC Plasma cell; Mat B Mature B cell.

and found the generation of IgE$^+$ PCs from B220$^+$CD19$^+$ B cells in the presence of CD4 T and NKT2 cells (Supplementary Fig. 6A–C). These data strongly suggest that thymic B cells differentiated into IgE$^+$ PCs in the thymus.

Recent reports showed that serum IgE levels are affected by the intestinal microbiome and dietary antigens[22]. To investigate their roles in developing thymic PCs, we analyzed them in germ-free (GF) and antigen-free (AF) mice, which are GF mice fed with amino-acid diets[23]. In these mice, there were no differences in total numbers or isotype compositions of thymic PCs, whereas the numbers of IgA- and IgG-producing PCs in the periphery were significantly decreased (Fig. 2F, G). Accordingly, serum IgE levels were not changed in GF and AF mice (Fig. 2H). These data indicate that intestinal antigens do not affect the development of thymic PCs, suggesting they produce natural antibodies.

**Thymic PCs are BM-derived tissue-resident cells.** Because thymic PCs are only a few thousand in their numbers, we excluded the possibility of blood contamination by measuring a positive fraction of intravenously injected anti-CD45.2 antibodies (ivCD45.2). We found that around 40–50% of splenic CD19$^+$IgD$^+$ mature B cells and PCs were stained with ivCD45.2, while thymic PCs were rarely positive, indicating they are not part of the circulating population (Supplementary Fig. 7A). To test the possibility of migration of peripheral PCs into the thymus, we sorted peripheral PCs and intravenously transferred them into congenic host. Compared to the spleen, donor-derived CD45.1$^+$ PCs were not detected in the thymus, suggesting it is unlikely that thymic PCs were migrated from peripheral tissues (Supplementary Fig. 7B). To further investigate whether thymic PCs are tissue-resident, we analyzed mice 2 weeks after parabiosis surgery. More than 80% of thymic B220$^{hi}$ B cells and CD138$^+$ PCs were host origin in each parabiont, indicating that they are tissue-resident cells (Fig. 3A).

The development of thymic PCs began within 1 week after birth, and their numbers were four times higher than those of BM in 7-week-old mice. Remarkably, no IgE$^+$ PCs in the BM were found while IgE$^+$ PCs were detected in the thymus, and their numbers increased with age (Fig. 3B). These early developmental kinetics suggest that they might derive from fetal precursors. To investigate this possibility, we transplanted the thymi of newborn BALB/c mice (CD45.1) underneath the kidney capsule of an adult congenic host (CD45.2) and analyzed B cell development (Fig. 3C). CD19$^+$ thymic B cells rarely differentiated from the donor thymocytes in these mice, indicating that they are not derived from fetal liver progenitors. Consistent with this, thymic PCs were replaced by donor cells with similar efficiency to the spleen in irradiation BM chimeras (Supplementary Fig. 7C). Overall, results from parabiosis experiments and newborn thymus graft indicate that thymic PCs are tissue-resident cells that develop from BM precursors after birth.

**scRNA-seq reveals the developmental landscape of thymic B cells.** To investigate the developmental landscape of thymic B cells at the clonal levels, we sorted equal numbers of CD138$^+$ PCs and B220$^+$ mature B cells from thymi of BALB/c mice (Supplementary Fig. 8A), followed by scRNA-seq paired with V(D)J sequencing. We profiled a total of 2053 quality control (QC)-positive thymic B cells from two pooled replicates, with an average of 1905 genes and 13,655 unique molecular identifiers (UMIs) per cell (Supplementary Fig. 8B–L). Of 2053 QC-positive cells, 71% of cells had contigs matched with both heavy and light chain sequences from paired V(D)J sequencing. We applied a graph-based clustering algorithm to identify nine clusters, which were organized into transitional B cells, mature B cells, memory B cells, plasmablasts (PBs), and PCs (Fig. 4A), based on the expression of B cell lineage markers including *Cd24a*, *Fcer2a*, *Ccr6*, *Prdm1*, and *Sdc1* (Fig. 4B, C). Antigen-experienced memory B cells express CCR6[24,25], and transitional B cells were identified based on their specific expression pattern of known signature genes of transitional B cells obtained from a public dataset[26] (Supplementary Fig. 9A). We found abundant *Vpreb3* transcripts in transitional B cells as reported previously[15] (Fig. 4B). However, VPREB3 proteins were undetectable in thymic and splenic B cells of adult mice (Supplementary Fig. 9B), and we also found that *Vpreb3* transcripts are highly detected in splenic transitional B cells when we analyzed a public dataset as described in the method (Supplementary Fig. 9C). Thus, mRNA expression of *Vpreb3* is not unique to pro/pre-B cells and is also expressed in transitional B cells. Among total PCs, five clusters expressing different isotypes were identified even though we excluded Ig genes for clustering (Fig. 4A and Supplementary Fig. 10A). IgE$^+$ PCs were further divided into two clusters (IgE1 and IgE2), mainly discriminated by MHC II-related genes (Fig. 4C and Supplementary Fig. 10B). With paired V(D)J sequencing, we detected 1,106 distinct clonotypes from 1312 PCs (including PBs), which used a diverse combination of V-J paring (Supplementary Fig. 10C). We found only 1–3% of PC clonotypes were repeated more than twice (Fig. 4D), and their diversity index was 0.99 (Fig. 4E), indicating that thymic PCs are extremely polyclonal. Next, we examined whether thymic PCs are generated after a GC reaction by analyzing the frequency of SHMs. We used constructed CDR1/2 sequences of V$_H$ genes obtained from transitional and mature B cells as complete germ-line BCR sequences are not available for the BALB/c mice we used for scRNA-seq analysis. As a result, we found 84% of thymic PCs were free from SHMs (Fig. 4F), whereas only 26% of GC B cells obtained from a public dataset[27] had no mutations at CDR1 or CDR2. We further analyzed the physico-chemical properties of CDR3 regions of each isotype using principal component analysis (PCA) of Kidera Factors[5] (Fig. 4G), which showed no difference between isotypes. Collectively, the above findings show that the mouse thymus contains B cells from the transitional B stage to PCs with a highly diverse repertoire without SMH.

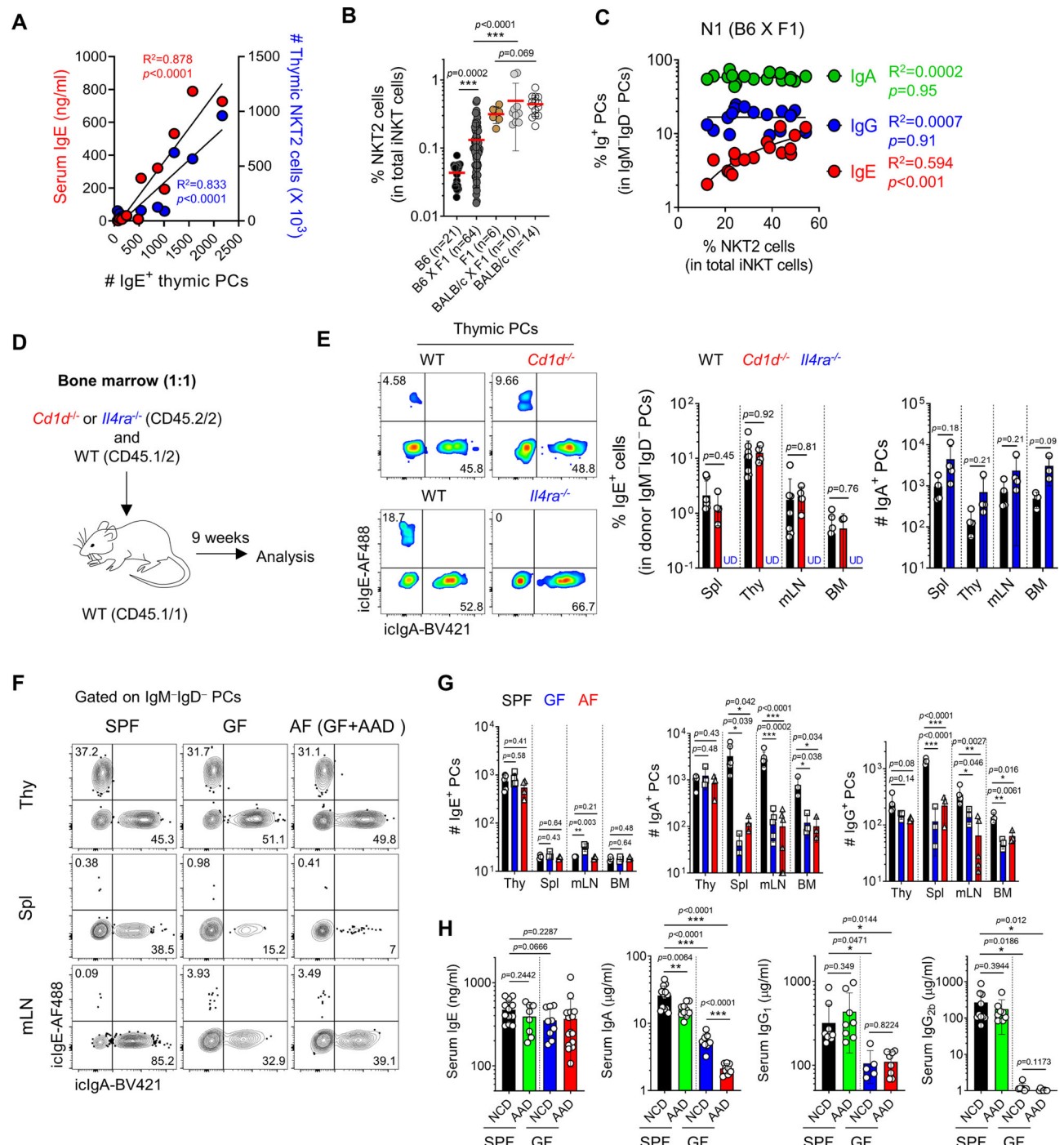

**Fig. 2 iNKT cells regulate the development of IgE⁺ thymic PCs. A** A total of 1–8-week-old BALB/c mice were analyzed for their serum IgE concentrations by ELISA and the numbers of thymic NKT2 and IgE⁺ PCs. Graph shows their correlations. **B** Graph shows frequencies of NKT2 cells among total iNKT cells in indicated mice. **C** Graph shows a correlation between frequencies of thymic NKT2 cells and PCs (N = 17) from F1 X B6 mice. **D** Schematic of an experimental strategy of mixed BM chimera. **E** Total thymocytes from BM chimera generated as in (**D**) were enriched for CD138⁺ cells by MACS. Representative plots show IgE⁺ thymic PCs derived from WT and *Cd1d⁻/⁻* (upper) or *Il4ra⁻/⁻* (lower) donors. Graph shows statistical analysis (N = 4–8). **F–H** SPF or GF BALB/c mice were fed with either NCD or AAD and analyzed for thymic PC development (N = 3–6). Representative FACS plots show icIgE, and icIgA expression of PCs in indicated tissues (**F**), and graphs show statistical analysis (**G**). Serum IgE, IgA, IgG₁, IgG₂ᵦ concentrations of indicated BALB/c mice were measured by ELISA (N = 8–15) (**H**). Results are from two independent experiments. Numbers indicate the frequencies of cells in each quadrant. Data are presented as mean values ± SD (**B**, **E**, **G**, and **H**). Each dot represents an individual mouse. Unpaired two-tailed *t*-tests were used. \*\*\**p* < 0.001, \*\**p* < 0.01, \**p* < 0.05. Not significant (*p* > 0.05). Linear regression was used to calculate goodness of fit (R²) and *P* values. PC Plasma cell; SPF Specific pathogen-free; GF Germ-free; AF Antigen-free; Thy Thymus; Spl Spleen; mLN Mesenteric lymph node; BM Bone marrow; NCD Normal chow diet; AAD Amino-acid diet; U.D. Undetected.

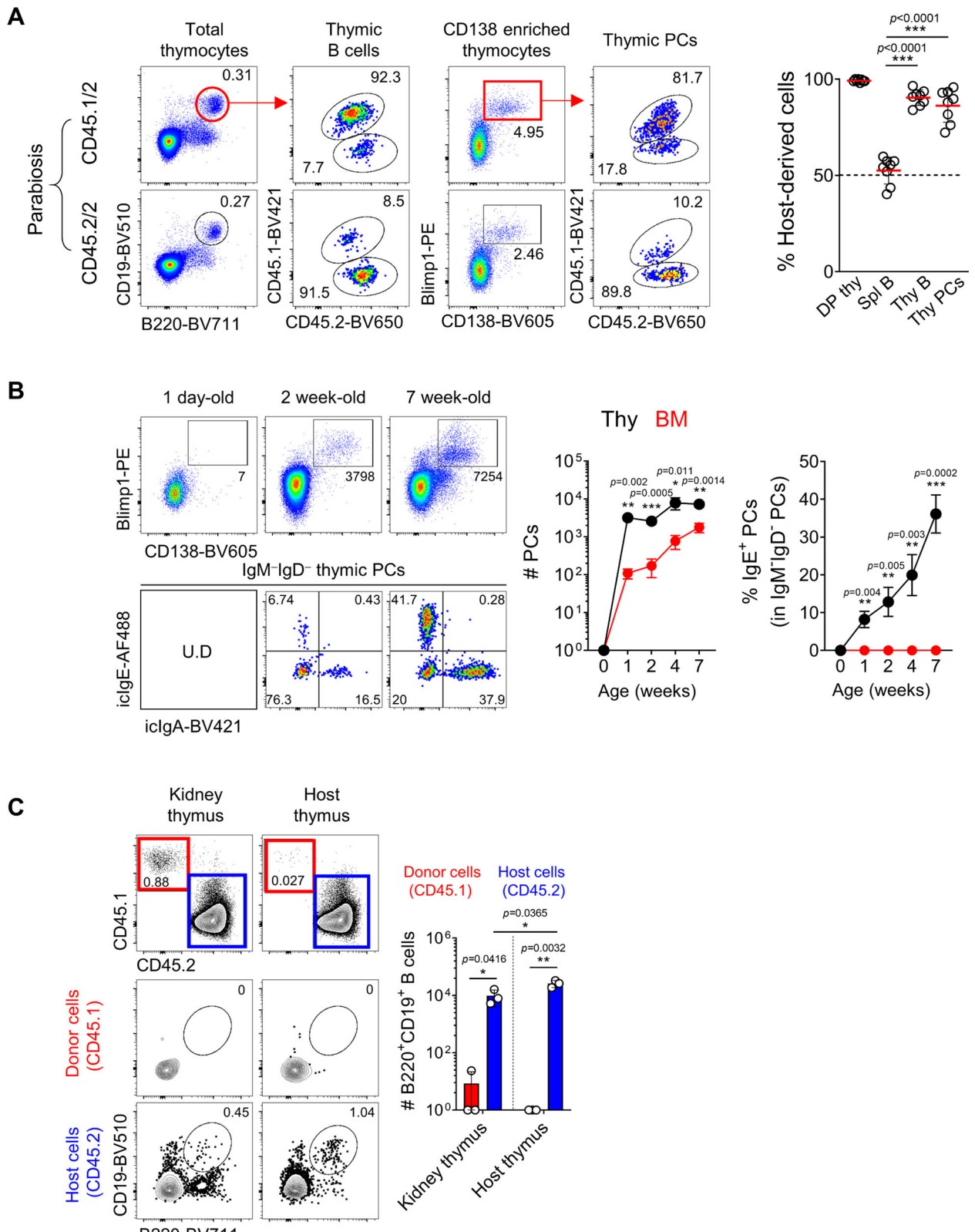

**Various stages of B cells develop in the thymus without GC reaction**. Based on scRNA-seq analysis, we next validated the presence of CD93$^+$ transitional B cells, IgD$^+$CD23$^+$ mature B cells, and CCR6$^+$ memory B cells among total B220$^+$CD19$^+$ thymic B cells by flow cytometry (Fig. 5A, B). PBs specifically expressed genes associated with cell proliferation (Fig. 4B, C), and we detected Ki67$^+$ cells among CD138$^+$Blimp1$^+$ cells (Fig. 5C).

Thus, we validated findings of scRNA-seq analysis, which showed various stages of developing B cells in the thymus.

scBCR repertoire analysis predicted thymic PCs develop without SHM, suggesting they are generated through a non-GC pathway. To confirm this, we analyzed $Cd4^{cre}Bcl6^{fl/fl}$ and $Icosl^{-/-}$ mice in which GC B cells do not develop[28]. Although these are B6 backgrounds and have a much-reduced number of IgE$^+$ PCs, IgA

**Fig. 3 Thymic PCs are BM-derived tissue-resident cells. A** Congenic pairs of CD45.1/2 and CD45.2/2 mice underwent parabiosis surgery and were analyzed after 2 weeks. Representative dot plots show chimeric ratios of B220$^+$CD19$^+$ B cells from total thymocytes or PCs enriched with CD138 antibody using MACS in the thymus (N = 8). Graph shows statistical analysis of host-derived cells in each parabiont. Numbers indicate frequencies of cells in adjacent gates. **B** Total thymocytes from mice of indicated ages were enriched for CD138$^+$ PCs by MACS, and representative dot plots are shown after gating out TCRβ$^+$ cells. Numbers indicate total cell numbers in adjacent gates (top) or frequencies of cells in each quadrant (bottom). Graph shows a statistical analysis of the total numbers of PCs (left) and frequencies of IgE$^+$ PCs (right) in the thymus and BM. **C** Recipient CD45.2/2 BALB/c mice were transplanted with thymic lobes of newborn CD45.1/1 mice underneath the kidney capsule and analyzed after 4 weeks. Representative dot plots show B220$^+$CD19$^+$ thymic B cells, and graph shows statistical comparison (N = 3). Numbers indicate frequencies of cells in adjacent gates. Data are presented as mean values ± SD (**A**, **B**, and **C**). Each dot represents an individual mouse. Unpaired two-tailed *t*-tests were used. \*\*\**p* < 0.001, \*\**p* < 0.01, \**p* < 0.05. DP Double positive; Mat B Mature B cells; PC Plasma cell; Thy Thymus; Spl Spleen; BM Bone marrow.

or IgG$^+$ PCs develop normally (Fig. 1C, D). In these mice, the numbers of total and class-switched PCs in the thymus remained unaffected, whereas those of spleens and PPs were significantly reduced (Fig. 5D, E), indicating PCs develop via the extra-follicular pathway in the thymus, at least as for IgA and IgG$^+$ ones.

**Thymic PCs express AIRE and locate in the medulla.** Previous reports showed that thymic B cells express AIRE and mediate negative selection of autoreactive CD4 T cells[16,17]. Based on this, we speculated that self-antigen mediated T-B cell interactions might promote PC differentiation expressing AIRE. As expected, TCR transgenic DO11.10 and OT-II mice and BCR transgenic MD4 mice had approximately 10-fold decreased numbers of thymic PCs, while they had similar numbers of peripheral B cells (Supplementary Fig. 11A, B). Accordingly, serum IgE levels of DO11.10 mice were significantly decreased (Supplementary Fig. 11C). We also found it is PCs that express AIRE among all CD19$^+$ thymic B cells (Supplementary Fig. 12A), although their expression levels were lower than those of medullary thymic epithelial cells (mTECs) (Supplementary Fig. 12B). In an imaging analysis, IgE$^+$EpCAM$^{lo}$Blimp1$^+$AIRE$^+$ PCs were detected in the thymic medulla, and they were distinct from EpCAM$^{hi}$AIRE$^+$ mTECs and PAX5$^+$B220$^+$ B cells (Supplementary Fig. 12C). We further confirmed that Blimp1$^+$AIRE$^+$ PCs do not express keratin5, a specific marker for mTEC (Supplementary Fig. 12D). Histocytometric analysis revealed that PCs are located more toward the medullary side than B220$^{hi}$ B cells, which are enriched at the cortico-medullary junction as shown previously[29] (Supplementary Fig. 12E).

mTECs express CD138 and Blimp1, albeit at low levels[30], and we showed thymic PCs express EpCAM and AIRE. To clarify the ambiguity between mTEC and PCs, we sorted EpCAM$^+$CD45.2$^-$ TECs, B220$^+$CD19$^+$ B cells, and CD19$^+$CD138$^+$ PCs and performed scRNA-seq (Supplementary Fig. 13A, B). Five clusters of thymic B cells were identified as in Fig. 4A, and mTECs were subdivided into three populations, as previously described[31]. Indeed, thymic PCs were transcriptionally distinct from them. Moreover, TECs did not express any surface or icIgs (Supplementary Fig. 13C), and thymic PCs expressed CD45, a marker of hematopoietic lineage (Supplementary Fig. 13D). In addition, thymic PCs had high levels of ER stress signature (GO:0034976), which is closely related to antibody production[32] (Supplementary Fig. 13E). Therefore, we concluded that thymic PCs residing in the medulla are different population from mTECs.

**The level of homeostatic IgEs correlates with the number of MCs in the gut and skin.** Survival and activation of MCs are enhanced by IgE bound to FcεRI receptors on their surface, independent of its antigen reactivity[33–35]. Based on this, we hypothesized that polyclonal thymic IgEs would promote the survival of MCs in the periphery. Indeed, in *Cd1d*$^{-/-}$ and

thymectomized BALB/c mice, the numbers of MCs in the gut were decreased by two to four-folds (Supplementary Fig. 14A). However, *Cd1d*$^{-/-}$ mice are deficient for peripheral iNKT cells, and thymectomized mice have potential defects in naïve T cell output. To overcome these issues, we generated mice specifically deleted with IgE-producing thymic PCs as depicted in Fig. 6A; we transplanted RAG1-deficient newborn thymi under the kidney capsule of WT BALB/c mice to support naïve T cell development and removed neck thymus 4 weeks later. Because iNKT cells mainly develop from fetal-derived thymocytes (Supplementary Fig. 15A, B), the transplanted RAG1-deficient thymi had much reduced iNKT cells and IgE$^+$ PCs than those of the neck thymus (Fig. 6B). Therefore, although not complete, these mice are largely free from issues related to the usage of *Cd1d*$^{-/-}$ or thymecto-mized mice. Remarkably, upon food anaphylaxis (FA) induction, these mice had an average of 2 °C (or 41.6%) fewer temperature drops compared to control mice (Fig. 6C). Consistent with this, these mice also had a 1.9-fold reduced serum MCPT-1 levels compared to control mice (Fig. 6D). Collectively, these results show the correlation between thymus-derived polyclonal IgEs, the number of MCs in the gut, and the severity of FA.

We additionally tested the effect of thymic IgEs in the passive cutaneous anaphylaxis (PCA) model, in which IgE from OVA-sensitized donor serum activates pre-existing MCs in the recipient skin. We observed a 12-fold reduction in MC numbers on average in the ear skin of *Cd1d*$^{-/-}$ mice (Supplementary Fig. 14B) and tested whether PCA responses are inhibited in *Cd1d*$^{-/-}$ recipients. As expected, when serum from immunized WT mice was transferred to *Cd1d*$^{-/-}$ hosts, there was an average 6.5 fold reduction in dye leakage in the skin (Fig. 6E, F). Together with results from the FA model, these data show the critical role of thymic IgEs in maintaining MC homeostasis in the periphery.

## Discussion

In this study, we unexpectedly found that thymus-resident CD138$^+$Blimp1$^+$ PCs are the source of homeostatic IgE in BALB/c mice. They originated from BM precursors from the early post-natal period and depended on IL-4 produced from iNKT cells. scRNA-seq analysis revealed that thymus supports multi-stages of B cell development from transitional B cells to PCs without SHM. Thymus-derived IgEs expanded MCs in the gut and skin, which correlated with the severity of anaphylactic responses. Overall, these results show the critical role of thymic PCs in maintaining serum IgE levels and MC homeostasis.

Previous studies showed that a small part of thymic B cells expresses AIRE and promiscuous self-antigens participating in the negative selection of autoreactive T cells[16,17]. However, the onto-geny of AIRE-expressing thymic B cells was not clear. In this report, we showed that thymic PCs express AIRE (Supplementary Fig. 12A, B). B220$^{hi}$ thymic B cells are enriched at the cortico-medullary junction[29], through which positively selected DP thymocytes migrate from the cortex to the medulla. Similar to DCs, autoreactive transitional or mature B cells would have a chance to uptake

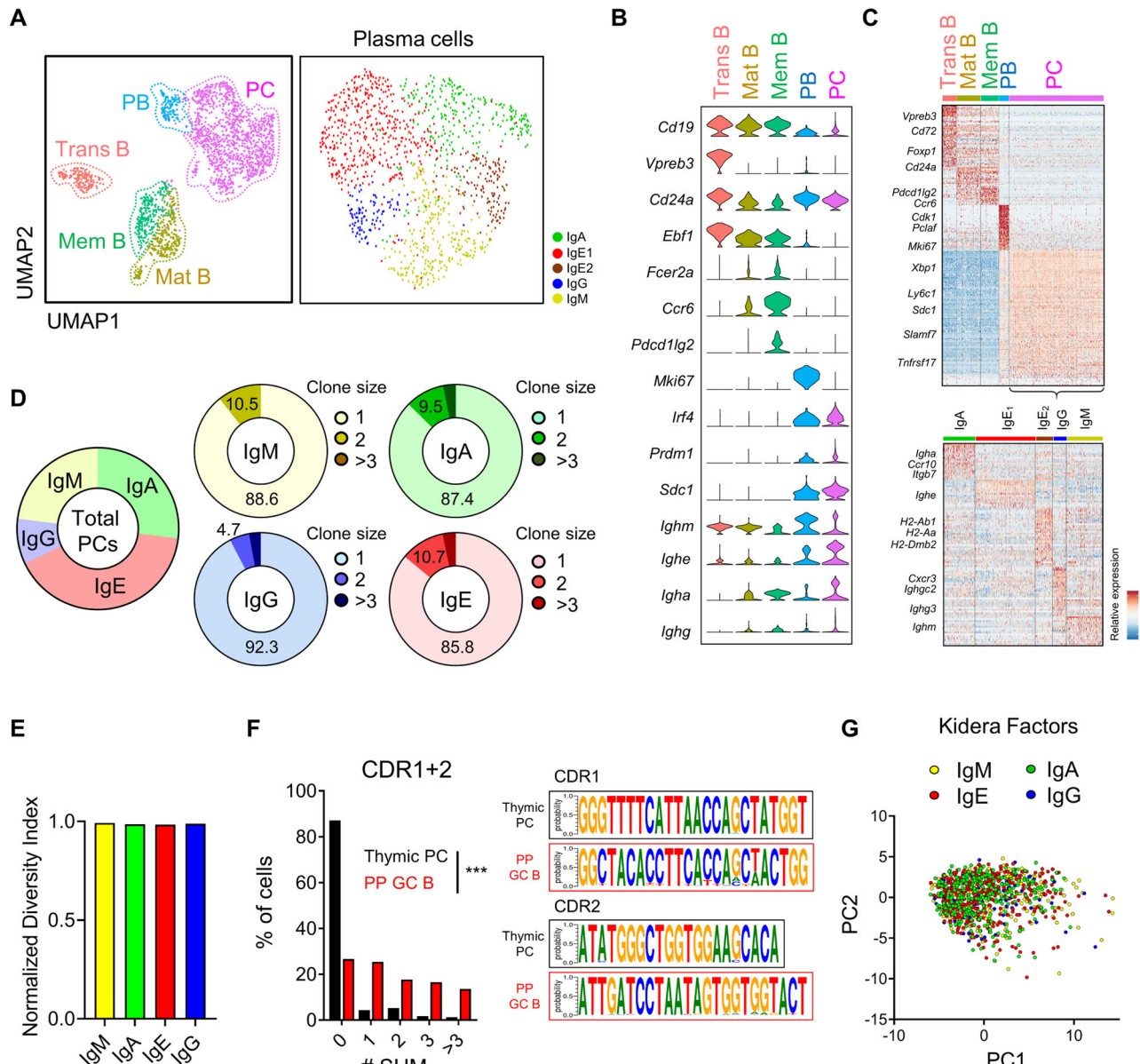

**Fig. 4 scRNA-seq reveals the developmental landscape of thymic B cells. A** Uniform manifold approximation and projection (UMAP) plot shows all cells (2053 cells, left) and PCs (1236 cells, right) from two pooled replicates. Each annotated B cell subtype and isotypes of PCs were labeled with indicated colors. **B** Violin plots show the expression levels of B cell subtype marker genes. **C** Heat-map shows the relative expression of differentially expressed genes of each annotated B cell subtype in all cells (top) and each isotype cluster in PCs (bottom). **D** Pie charts show the percentage of cells belonging to each isotype in PB/PCs (left) and the percentage of clone size per isotype (right). **E** Bar plot shows the Shannon equitability indices of clonotypes across isotypes in PB/PCs. **F** Graph shows the percentage of cells with different numbers of SHMs in the CDR1 and CDR2 sequences of heavy chains between thymic PCs and PP GC B cells (left). Relative CDR1 and CDR2 sequences for the most abundant V gene of the heavy chain were shown by using WebLogo (right). **G** PCA plot shows cells colored by different isotypes using Kidera Factors of heavy chain CDR3 regions. Two-sided chi-squared test was used in (**F**). ***$p < 0.001$. Trans B Transitional B cells; Mat B Mature B cells; mem B Memory B cells; PB Plasmablast; PC Plasma cell; SHM Somatic hypermutation; PP Peyer's patch; GC B Germinal center B cells.

self-antigens in the thymus, which would facilitate interaction with cognate autoreactive CD4 T cells[36]. Upon cognate interaction, thymic B cells are activated and undergo CSR in an activation-induced deaminase (AID)-dependent manner without GC formation. Supporting this hypothesis, a recent report showed CSR occurs before the formation of GCs[37]. Parabiosis experiments showed that about 20% of thymic PCs originated from circulation (Fig. 3A), and a similar portion of thymic PCs had the same sequence of CDR1 and CDR2 (Fig. 4F). Thus, some fraction of PCs would also migrate from the periphery. Other reports also showed that AIRE is important for thymic regulatory T cell (Treg) development[38], and

thymic B cells participate in the positive selection of Tregs[39]. Therefore, it is possible that AIRE⁺ thymic PCs not only participate in the negative selection of autoreactive T cells but also in the positive selection of Tregs. From this perspective, thymic PCs would produce autoreactive antibodies, but it is unlikely that these antibodies would induce autoimmune disease as their affinity is low, and cognate CD4 T cells are eliminated while Tregs are selected. However, the exact mechanisms of T cell tolerance by thymic PCs require further investigations.

In mice, systemic anaphylaxis is mediated by two distinct mechanisms: a classic pathway involving IgE-FcεRI and MCs and

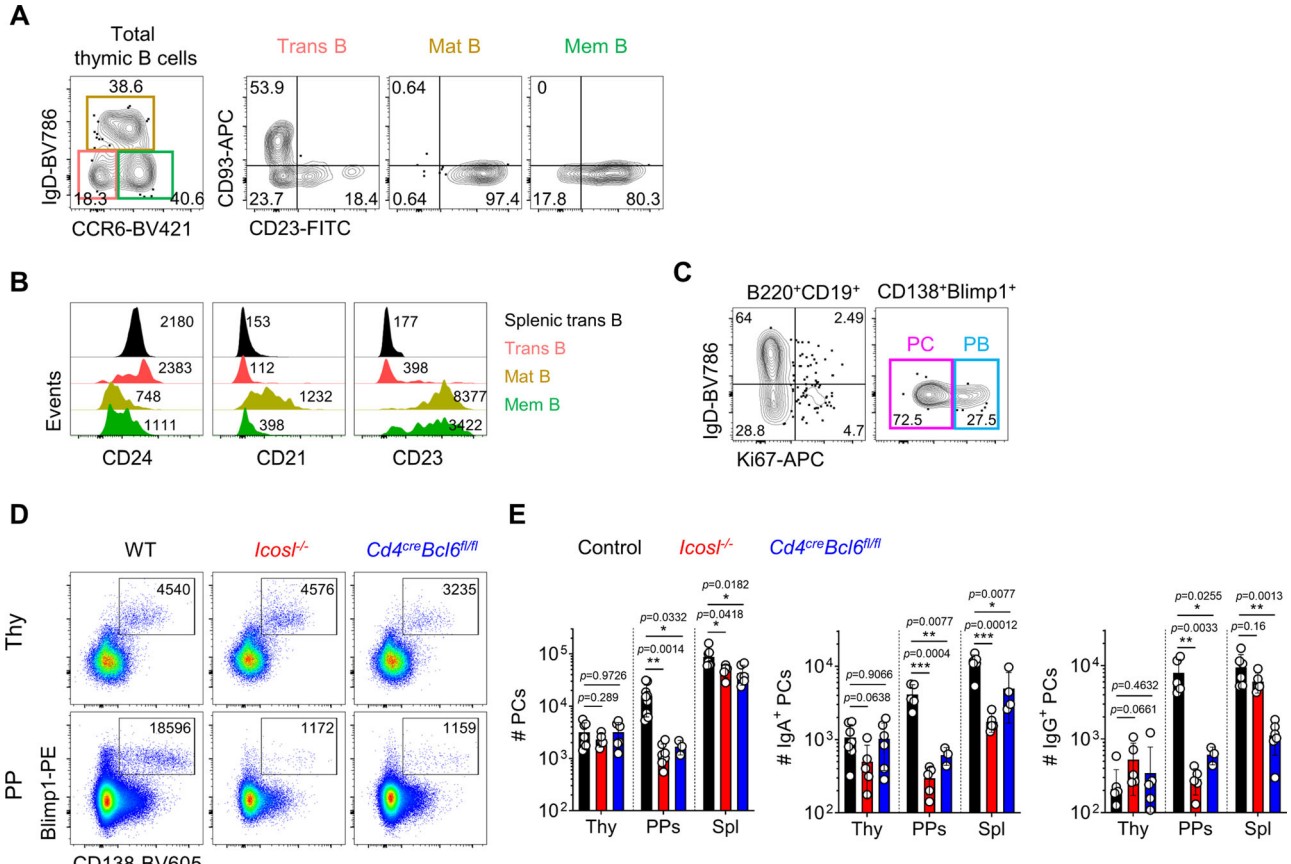

**Fig. 5 Various stages of B cells develop in the thymus without GC reaction. A** Representative FACS plots show IgD⁻CCR6⁻ transitional B, IgD⁺CCR6^lo mature B, and IgD⁻CCR6^hi memory B cells from total B220⁺CD19⁺ thymic B cells (left). Expression levels of CD93 and CD23 from indicated cells were shown (right three panels). Numbers indicate frequencies of cells in adjacent gates. **B** The histogram shows MFI of CD24, CD21, and CD23 expression from indicated cells. Numbers indicate MFI value. **C** Representative FACS plots show Ki67 expression in B220⁺CD19⁺ B cells (left) and CD138⁺Blimp1⁺ PCs (right) from total and CD138 enriched thymocytes, respectively. **D**, **E** Representative dot plots show PCs from CD138 enriched thymocytes (upper) and total cells in PPs (lower) from indicated mice (**D**). The number of cells was shown in adjacent gates. Results are from three independent experiments. Graphs show statistical analysis (N = 3–9 for each group) (**E**). Data are presented as mean values ± SD. Each dot represents an individual mouse. Unpaired two-tailed t-tests were used for comparison. \*\*\*p < 0.001, \*\*p < 0.01, \*p < 0.05. Not significant (p > 0.05). PB Plasmablast; PC Plasma cell; Thy Thymus; Spl Spleen; PPs Peyer's patches; mLN Mesenteric lymph node; BM Bone marrow.

an alternative pathway involving IgG-FcγRs and macrophages[40]. Previous reports showed that the FA model using OVA-alum is more likely to depend on the latter as IgG blockage, but not IgE, inhibited FA[41]. A recent report showed that ILC2 in the gut expands by IL-33 derived from atopic skin and secrete IL-4, which induces the proliferation of MCs[42]. In this report, expanded MCs promoted FA in an IgE-independent manner, suggesting the number of pre-existing MCs could be a critical risk factor of FA. Because IgE-deficient mice show a dramatic reduction of MCs in the gut[43], we speculated that thymic IgE would promote FA by increasing the number of MCs in the gut and skin before antigenic challenge. Consistent with this, we showed the correlations between the numbers of MCs, serum IgE levels, and thymic IgE producing PCs in Cd1d⁻/⁻ and thymectomized BALB/c mice (Fig. 1 and Supplementary Fig. 14). To show this more convincingly, we also generated a mice model specifically deleted with thymic IgE producing PCs (Fig. 6A), as a genetically engineered system that specifically deletes thymic IgEs is not currently available.

In humans, thymic B cells with a distinct phenotype also expressed AIRE[44]. Another study showed human thymic PCs are found from the first year of life and reside in the perivascular space secreting Igs specific to viral antigens[45]. More recently, they showed that various developmental stages of B cells are present in

the human thymus, including CD138⁺ PCs producing all classes of natural antibodies[46]. Analogous to these human data, we showed thymic B cells from transitional B cells to plasma cells also develop in mice. Instead of iNKT cells, humans have IL-4-secreting PLZF^hi T cells[47,48], which could be the source of innate IL-4 to determine the development of IgE-producing PCs. In the future, it is crucial to understand the role of thymic PCs in humans by analyzing the relationship between the frequencies of IgE-producing thymic PCs and atopic or allergic tendencies. Overall, the results of this study provide new insights into our understanding of the genetic influence of allergic disorders determined by iNKT cells and thymic PCs.

## Methods

**Mice**. CD45.1 (B6.SJL-Ptprc^a Pepc^b/BoyJ) and WT B6 mice, and Rag1⁻/⁻ (C.129S7(B6)-Rag1^tm1Mom/J), Cd1d⁻/⁻ (C.129S2-Cd1tm1Gru/J), Il4ra⁻/⁻ (BALB/c-Il4ratm1Sz/J), CD45.1 (CByJ.SJL(B6)-Ptprc^a/J), DO11.10 (C.Cg-Tg(DO11.10)10Dlo/J), and WT BALB/c (BALB/cByJ or BALB/cJ) mice were purchased from the Jackson Laboratory. BALB/cAnNCrl mice were purchased from Charles River. In all experiments, substrain of WT and mutant BALB/c mice were matched together for comparison. OT-II (B6.Cg-Tg (TcraTcrb)425Cbn/J), Icosl⁻/⁻ (B6.129P2-Icosl^tm1Mak/J), CD4^Cre (B6.Cg-Tg(Cd4-cre)1Cwi/BfluJ), and Bcl6^fl/fl (B6.129 S(FVB)-Bcl6^tm1.1Dent/J) mice were provided by Sin-Hyeog Im (POSTECH, Korea), and MD4 (C57BL/6-Tg (IghelMD4)4Ccg/J) mice were from Yoontae Lee (POSTECH, Korea). Zsgreen-Tbx21 and KN2 mice were previously described[12]. GF and AF BALB/c mice were maintained as described previously[23]. All the mice

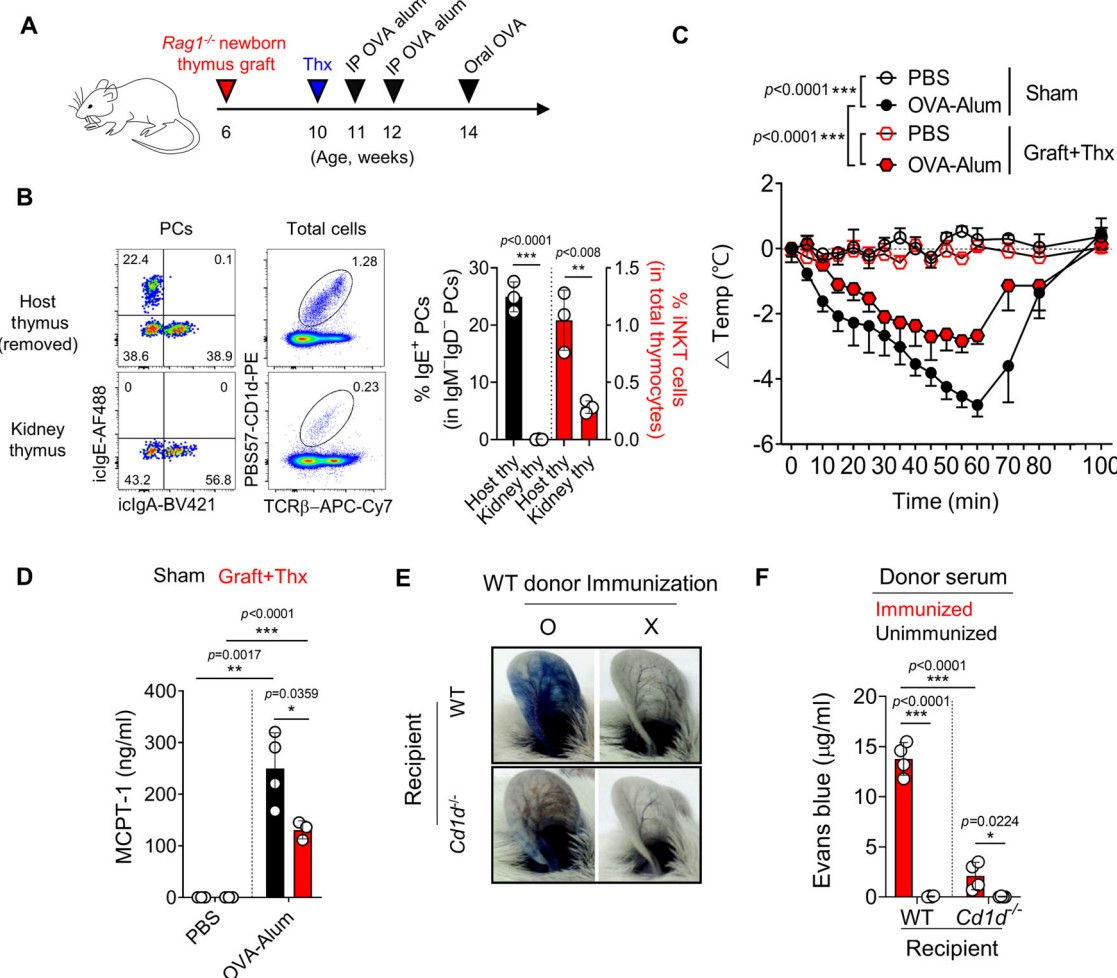

**Fig. 6 The level of homeostatic IgEs correlates with the severity of food anaphylaxis. A–D** The figure shows experimental schematics for food anaphylaxis (FA) induction in *Rag1*^−/−^ thymus-grafted and thymectomized BALB/ mice (**A**). Representative dot plots show IgE⁺ thymic PCs (left) and iNKT cells (right) in mice grafted with RAG1-deficient thymus (**B**). Numbers indicate frequencies of cells in adjacent gates. Graph shows body temperature changes in the indicated group (N = 3–4 for each group) (**C**). Graphs show serum concentrations of MCPT-1 measured by ELISA in indicated mice (**D**). **E, F** Twenty microliter of sera from NP-OVA immunized or unimmunized WT BALB/c mice were intradermally injected into ear pinnae of naïve WT and *Cd1d*^−/−^ recipients. Recipient mice were intravenously injected with NP-OVA and Evans blue 24 h later. Pictures show ear pinnae from indicated mice (**E**), and graph shows concentrations of Evans blue dye in harvested ear pinnae (N = 4 for each group) (**F**). Data are presented as mean values ± SD (**B, D,** and **F**). Each dot represents an individual mouse. Unpaired two-tailed *t*-tests (**B, D,** and **F**) and two-way ANOVA (**C**) were used. ***p < 0.001, **p < 0.01, *p < 0.05. Not significant (p > 0.05). Thx Thymectomy; siLP Small intestinal lamina propria; MCPT-1 Mast cell protease-1.

are maintained in the animal facility of Pohang University of Science and Technology (POSTECH) Biotech Center in accordance with the Institutional Animal Care and Use Committee of POSTECH.

**Flow cytometry.** TECs were isolated as previously described[31]. Briefly, thymus was minced followed by incubating in RPMI1640 media with 0.5 mg/ml of Collagenase D (Roche) and DNase I (Sigma-Aldrich) for 45 min at 37 °C. For intracellular staining, an eBioscience Foxp3 staining buffer set was used. PBS-57 loaded or unloaded CD1d monomers were obtained from the tetramer facility of the US National Institutes of Health. Cells were analyzed on an LSR Fortessa II (BD Biosciences), and data were processed with FlowJo software. Following antibodies were used: Anti-CD3ε (PE, Invitrogen, 1:200) Anti-CD4 (BUV395, BD, 1:400), Anti-CD4 (PE, Invitrogen, 1:400), Anti-CD4 (APC-Cy7, Invitrogen, 1:200), Anti-CD5 (PerCP-Cy5.5, Biolegend, 1:400), Anti-CD8α (BV650, BD, 1:400), Anti-CD11b (PE, eBioscience, 1:400), Anti-CD11b (PerCP-Cy5.5, BD, 1:1000), Anti-CD19 (BV510, BD,1:200), Anti-CD19 (PE-Cy7, BD, 1:200), Anti-CD21 (PE-Cy7, eBioscience, 1:400), Anti-CD23 (FITC, eBioscience, 1:400), Anti-CD24 (BV605, Biolegend, 1:1000), Anti-CD38 (AF700, Invitrogen, 1:400), Anti-CD43 (PE-Cy7, Biolegend, 1:1000), Anti-CD44 (redFluor710, TONBO, 1:400), Anti-CD45.1 (PB, Biolegend, 1:200), Anti-CD45.1 (AF700, BD, 1:200), Anti-CD45.2 (BV650, Biolegend, 1:200), Anti-CD45.2 (PerCP-Cy5.5, Invitrogen, 1:200), Anti-CD45.2 (FITC, Invitrogen, 1:200), Anti-CD45R/B220 (BV711, BD, 1:1000), Anti-CD45R/B220 (APC, Invitrogen, 1:400), Anti-CD45R/B220 (FITC, TONBO, 1:400), Anti-CD49b

(PB, Biolegend, 1:200), Anti-CD80 (PE, eBioscience, 1:100), Anti-CD86 (FITC, eBioscience, 1:100), Anti-CD93 (APC, eBioscience, 1:200), Anti-CD117 (ckit) (PE-Cy7, Invitrogen, 1:1000), Anti-CD138 (BV605, BD, 1:200), Anti-CD138 (Biotin, BD, 1:200), Anti-CD170 (Siglec-f) (PE, BD, 1:1000), Anti-CD183 (CXCR3) (PE-Cy7, Biolegend, 1:100), Anti-CD185 (CXCR5) (BV421, BD, 1:50), Anti-CD196 (CCR6) (BV421, Biolegend, 1:200), Anti-CD326 (EpCAM) (FITC, Biolegend, 1:200), Anti-TCRβ (APC-Cy7, BD, 1:200), Anti-FcεRI (APC, Invitrogen, 1:100), Anti-MHC-II (APC, eBiosciecne, 1:4000), Anti-Ly6C (eFluor450, Invitrogen, 1:400), Anti-VPREB3 (PE, Biolegend, 1:100), Anti-EOMES (eFluor450, Thermo Fisher Scientific, 1:100), Anti-EOMES (AF488, eBiosciecne, 1:200), Anti-IFN-gamma (PE-CF594, BD, 1:400), Anti-IL-4 (BV421, Biolegend, 1:100), Anti-Ki67 (FITC, Thermo Fisher Scientific, 1:200), Anti-IL-17A (BV650, BD, 1:400), Anti-PLZF (AF647, BD, 1:400), Anti-RORγt (PerCP-Cy5.5, BD, 1:400), Anti-RORγt (PETR, BD, 1:400), Anti-Tbet (PE-Cy7, Thermo Fisher Scientific, 1:400) Anti-Blimp1 (PE, Invitrogen, 1:100), Anti-PAX5 (APC, eBioscience, 1:400), Anti-IRF4 (PE, eBioscience, 1:200), Anti-IgM (PerCP-Cy5.5, BD, 1:200), Anti-IgD (BV786, BD, 1:2000), Anti-IgA (BV421, BD, 1:400 for surface staining and 1:4000 for intracellular staining), Anti-IgE (AF488, Biolegend, 1:400 for surface staining and 1:4000 for intracellular staining), Anti-IgE (FITC, BD, 1:400 for surface staining and 1:4000 for intracellular staining), Anti-IgG(Fc) (AF647, SouthernBiotech, 1:400 for surface staining and 1:10000 for intracellular staining), Anti-IgG1 (PETR, Jackson ImmunoResearch, 1:1000 for intracellular staining), Anti-AIRE (eFluor660, Invitrogen, 1:400), Anti-human CD2 (FITC, BD, 1:50)

**ELISA**. Serum concentrations of IgE (Biolegend, #432404), IgM (Thermo Fisher Scientific, #88-50470-88), IgA (Thermo Fisher Scientific, #88-50450-86), IgG1 (Thermo Fisher Scientific, #88-50410-86) IgG2a (Thermo Fisher Scientific, #88-50420-22), and MCPT-1 (Thermo Fisher Scientific, #88-7503-22) were measured by ELISA kits. ProcartaPlex Immunoassays (Thermo Fisher Scientific, #EPX070-20815-901) was used for measuring serum Igs in Supplementary Fig. 2A according to the manufacturer's instructions.

**Single-cell RNA sequencing**. Sorted B cells and mTEC were loaded on Chromium Single Cell G Chip kit (PN-1000127, 10X Genomics) to capture 2000~ 4000 cells, and libraries were generated according to the manufacturer's instructions (10X Genomics). Libraries were sequenced on the Illumina HiSeq X (paired-end 100 bp reads), aiming at an average of 50,000 read pairs per cell for scRNA-seq or 5000 read pairs per cell for paired V(D)J sequencing.

**scRNA-seq data analysis**. Raw reads for scRNA-seq were mapped to the mouse reference genome (GRCm38) using the Cell Ranger software (v3.1.0) with the Ensembl GRCm38.99 GTF file. For each pooled replicate, a gene-by-cell UMI count matrix was generated with default parameters except for expect-cells = 4000 or 2000. Using the emptyDrops function of the DropletUtils (v1.61) R package[49] with FDR < 0.05, empty droplets were removed. To filter out low-quality cells, cells with more than 10% of UMIs assigned to mitochondrial genes and less than 1000 total UMI count were excluded using the calculateQCMetrics function of the scater (v1.14.6) R package. The aggregated UMI count matrix across replicates was normalized by dividing raw UMI counts by cell-specific size factors calculated by the computeSumFactors function of the scran (v.1.14.6) R package[50]. The normalized counts were log2-transformed with a pseudocount of 1. Highly variable genes (HVGs) were defined as top 1000 genes with respect to biological variability estimated by the modelGeneVar function of the scran package. All QC-positive cells were grouped into 16 clusters using the FindClusters function of the Seurat (v3.1.5) R package[51] with the first 20 principal components (PCs) of HVGs and resolution = 0.8, and visualized in the two-dimensional UMAP plot using the RunUMAP function of the Seurat package with the same PCs. Each cell cluster was annotated based on immune cell markers, and non-B cell clusters (clusters 14 and 15) were removed (Supplementary Fig. 8B, C). After removing non-B cell clusters, cells were grouped into 11 clusters using the same method as above except for using HVGs excluding Ig genes and 15 PCs. Cluster 8 with high mitochondrial proportion was excluded (Supplementary Fig. 8D, E). After removing cluster 8, cells were re-grouped into 12 clusters using the same method as above except for 25 PCs. We removed cluster 9 that had a low number of detected genes per cell (Supplementary Fig. 8F, G). Batch effects between two pooled replicates were corrected by using the RunHarmony function of the harmony (v1.0) R package[52] on 20 PCs of HVGs excluding Ig genes. The corrected PCs were used as an input of the clustering and visualization method described above. The 9 clusters were grouped into five B cell subtypes (transitional B, mature B, memory B, plasmablasts, and plasma cells) based on the expression of B cell lineage markers. For each B cell subtype, subtype-specific marker genes were identified using the FindAllMarkers function of the Seurat package with default parameters. PC clusters were further subdivided into 8 clusters and visualized in the UMAP plot using the same method as above except for 13 PCs. For the scRNA-seq analysis of mTECs, all QC-positive cells were firstly grouped into 14 clusters using the FindClusters function with the first 15 PCs of HVGs and resolution = 0.8. After removing clusters (clusters 7 and 8) annotated as T cells, cells were re-grouped into 12 clusters using the same method as above. For a public scRNA-seq dataset of splenocyte (Splenocytes from C57BL/6 mice (v1.1, v2), Single Cell Immune Profiling Dataset by Cell Ranger 4.0.0, 10x Genomics, (2020, August 25), all QC-positive cells were grouped into 20 clusters using the FindClusters function with the first 15 PCs of HVGs and resolution = 0.8. Cells annotated as B cells were subgrouped into 11 clusters using the same method as above except for 20 PCs.

**BCR repertoire analysis**. Raw reads for paired V(D)J sequencing were processed using the cellranger vdj of the Cell Ranger (v3.1.0). The V(D)J segment-based reference was constructed using the cellranger mkvdjref command of the Cell Ranger by downloading mouse V(D)J segment sequences from IMGT with fetch-imgt. We considered only contigs called as both productive and high-confidence for further analysis. A set of cells was grouped into the same clonotype if they have the same V/J composition and identical CDR3 sequences in both heavy and light chains. To calculate the frequency of SHMs within thymic PCs, we constructed the germ-line CDR1/2 sequences of Vh genes from transitional and mature B cells by using MakeDB of the Change-O (v1.0.0) python package[53]. The SHMs of thymic PCs were identified by comparing them with thymic B220+ B cells using the seqDist function of the alakazam (v1.0.2) R package[53]. The frequency of SHMs within GC B cells from a public dataset was calculated by constructing germ-line CDR1/2 sequences from non-GC B cells and comparing them with those of GC B cells. CDR1/2 logo sequences were visualized by using WebLogo (v.3.7.4)[54]. The normalized diversity index was calculated using the ComputeShannonIndex function as previously described[55]. Kidera Factors of CDR3 amino-acid sequences of heavy chains were calculated using the Kidera Factors function of the Peptides (v2.4.2) R package[56] and visualized in the PCA plot using the prcomp function in R

based on the cell-by-cell Minkowski distance of order 4 matrix. Circos plots of Vh-Jh pairs were visualized using the chordDiagram function of the circlize (v0.4.11) R package[57].

**CD4 T cell depletion**. Anti-CD4 depleting antibody (100 μg; GK1.5, BioXcell) was administered to 2.5 weeks-old BALB/c mice every 3 days for 4 weeks.

**MACS enrichment**. Samples were stained with biotinylated anti-CD138 (281-2; BD) antibody and incubated with streptavidin microbeads according to the manufacturer's instructions (Miltenyi).

**Immunofluorescence**. Tissue imaging and histocytometric analysis were performed as previously described[13]. Briefly, thymic tissues were fixed with 4% paraformaldehyde for 1 h and snap-frozen. Twenty-micrometer tissue sections were blocked with 5% bovine serum albumin for 1 h and stained with indicated markers at RT. Leica DM6B was used for imaging, and ImageJ was used for image analysis. Following antibodies were used: Anti-CD45R/B220 (BV480, BD, 1:200), Anti-PAX5 (AF488, BD, 1:200), Anti-Cytokeratin5 (purified, Abcam, 1:400), Anti-CD326 (EpCAM) (FITC, Biolegend, 1:200), Anti-Blimp1 (PE, Invitrogen, 1:100), Anti-IgE (AF488, Biolegend, 1:200), Anti-AIRE (eFluor660, Invitrogen, 1:200), Anti-PE antibody (Novusbiologicals, 1:200), Anti-Rabbit IgG (AF488, Biolegend, 1:1000), Anti-Rabbit IgG (AF555, Invitrogen, 1:1000).

**Mixed BM chimeras**. BM was prepared from the femurs and tibiae of mice, and depleted of mature T cells using anti-CD4 and anti-CD8 microbeads (Miltenyi). Recipient mice were lethally irradiated (400 rad X 2) and received $5 \times 10^6$ donor BM cells. Chimeras were analyzed 9–10 weeks after transplantation.

**Thymectomy**. Thymectomy was performed as previously described[58]. Briefly, 6-week-old mice were anesthetized by intraperitoneal injection of 2.5% avertin solution (Sigma), and the upper part of the thorax was opened, and a small opening was made in the pleura (1-to 2- cm incision). Then, thymic lobi were carefully removed onto the exterior surface with forceps, and the skin was closed with a puncher.

**Parabiosis**. A total of 6 to 7-week-old congenic CD45.1/2 and CD45.2/2 mice were surgically connected in parabiosis as previously described[59]. After corresponding lateral skin incisions were made from elbow to knee in each mouse, forelimbs and hindlimbs were tied together with suture (VICRYL), and the skin incisions were closed using Autoclips (BD). After surgery, mice were maintained on a diet and drinking water supplemented with 2 mg/ml of sulfamethoxazole (sigma) and 0.4 mg/ml of trimethoprim (sigma) to prevent infection.

**Thymus transplantation**. As previously described[60], thymi were harvested from newborn CD45.1/1 BALB/c mice. An incision was made on the left abdomen of anesthetized 6-week-old CD45.2/2 BALB/c mice, and the donor thymic lobe was grafted underneath the kidney capsule. The muscle layer was sutured, and the skin was closed using Autoclips (BD).

**Cell sorting and real-time qPCR**. MACS enriched thymocytes were sorted using MoFlo Astrios (Beckman Coulter), and cells with more than 95% purity were used for experiments. TRIzol (Invitrogen) and Quantitect reverse transcription kit (QIAGEN) were used for RNA isolation and cDNA synthesis. SYBR green (TAKARA) and a Viia7 (Life technologies) were used for amplification and detection. HPRT amplification was used for normalization in all real-time qPCR analysis. Primers are as follows: *Hprt* forward, 5′-CAGACTGAAGAGCTACT GTAATGATCA-3′ and reverse, 5′-TCA CAATCAAGACATTCTTTCCA-3′. Primers for switched and mature IgE transcripts are previously reported[9]. ε Switched transcript forward, 5′-CTCTGGCCCTGCTTATTGTTG-3′ and reverse, 5′- AGTTCACAGTGCTCATGTTCAG-3′; ε Mature transcript forward, 5′-TACG ACGAGAACGGGTTTGCTTAC-3′ and reverse, 5′-AGTTCACAGTGCTCAT GTTCAG-3′.

**Fetal thymus organ culture (FTOC)**. As shown previously[55], fetal thymuses on embryonic day 15.5 (E15.5) from BALB/c mice were isolated and cultured on hydrophilic isopore membrane filter (0.8 μm pore size, Millipore, ATTP01300) placed on gelfoam sponge (Millipore, Medford, MA) in RPMI 1640 media with 2′-deoxyguanosine (Bio Basic, Amherst, NY) for 7 days. Sorted cells were colonized using hanging drop culture and analyzed after 7 days.

**Passive cutaneous anaphylaxis**. As previously described[61], ~ 20 μl of serum from immunized and boosted mice were injected into the ear pinnae of recipients. After 24 h, the recipient mice were i.v. injected with 100 μg NP-OVA (LGC Biosearch Technologies) with 1% Evans blue. Ear pinnae were analyzed 30 min later for vascular leakage.

**Food anaphylaxis (FA)**. As previously shown[42], 6–7 week-old mice were injected with 100 μg of OVA (Sigma) and 2 mg of alum (Invivogen) at days 0 and 7. On day 21, mice were intragastrically administrated with 100 mg of OVA. Temperatures were measured every 5 min for 100 min using a TCAT-2LV animal temperature controller (Physitemp).

**Statistical analysis**. Prism software (GraphPad) was used for statistical analysis. Unpaired or paired two-tailed $t$-test, linear regression, one-way or two-way ANOVA, and two-sided chi-squared test were used for data analysis and the generation of $p$ or $r^2$ values.

**Reporting summary**. Further information on research design is available in the Nature Research Reporting Summary linked to this article.

## Data availability

The authors declare that all data that support our findings in this study are included in the supplemental information and available upon reasonable requests to the corresponding authors. Source Data are provided with this paper. The single-cell RNA and BCR sequencing data in this study have been deposited in the SRA database under the accession code PRJNA681739 and are available. The processed single cell data and source data are available at https://github.com/CB-postech/NATURE-COMMUNICATIONS-thymus-plasmcells. Source data are provided with this paper.

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

## Acknowledgements

This research has been supported by the POSCO Science Fellowship of POSCO TJ Park Foundation (to Y.J.L.), and by the Korean Ministry of Science, Information/Communication Technology and Future Planning (2022R1A2C1007692, 2021R1A4A1031754 to Y.J.L. and 2017M3C7A1048448, 2017M3A9B6073099, 2020R1A2C400163011, 2018R1A5A1025511 to J.K.K.). D.K. was funded by BK21 Plus (10Z20130012243) by the Ministry of Education, Korea.

## Author contributions

D.K. designed and performed experiments, analyzed data and wrote draft; E.S.P. and Y.H.C. analyzed scRNA-seq data; M.K. and M.S.L. performed imaging analysis; S.J. performed FA experiment; Y.K. and S.J. performed thymectomy; M.L. performed FTOC; S.L. provided research interpretation; J.K.K. and Y.J.L. analyzed data, and wrote the manuscript. Y.J.L. conceptualized the research.

## Competing interests

The authors declare no competing interests.
