## [Peer Review File · Nature Communications]

Homeostatic serum IgE is secreted by plasma cells in the thymus and enhances mast cell survivalREVIEWER COMMENTS

Reviewer #1 (Remarks to the Author):

Reports that plasma cells develop IgE that produce IgE. They also show that thymus derived IgEs increase mast cell numbers in peripheral tissues such as gut, which then are responsible for food allergy and anaphylaxis. These IgE producing B cells express Aire, which plays a role in deleting self reactive CD4 SP thymocytes.

Do the authors have direct evidence that IgE+ B-cells develop intrathymically, or do they migrate into the thymus from peripheral tissues? Can the authors show these cells develop in a 'closed system', such as thymus organ culture?

A central question not addressed is why is the thymus a specialised place that fosters these IgE producing B-cells? The argument given is that this is connected to the availability of IL4 producing NKT cells in thymus compared to other places. This argument is not strongly supported by the data. IL4+ NKT2 are in fact abundant in other sites, so why the need for a thymic specific population? Also, Blimp1+CD138+ B-cells are still present at reasonable numbers in NKT-deficient Cd1d KO mice, so the authors need to consider additional intrathymic cellular sources of IL4. Another key experiment here is to add back IL4 into Cd1d KO mice via IV injection – does this restore intrathymic PCs?

The thymus transplant experiments shown in Fig 3C are confusing. The conclusion is that thymic PCs are tissue resident cell that develop from BM precursors immediately after birth. The data shown suggests that thymus grafts from 1 week old mice do not contain donor derived thymic PCs. Data is shown for 4 weeks, but what happens if these grafts are harvested earlier? Surely to support the conclusion drawn, the authors would see thymic PCs in these grafts, which contain BM derived progenitors? Where is the data in this experiment that shows these cells are tissue resident?

The data shown in Fig 6, that thymic PCs express Aire, does not seem to have anything to do with the rest of the study. Do the authors think that their expression of Aire is important for their development/function, this could be done through analysis of Aire KO mice. Regarding the idea that thymic PCs express Aire, the authors try to demonstrate that these cells belong to the B-cell lineage and not TEC lineage. This is very important because both cell types are thought to express a common range of markers. The microscopy is not particularly clear – can the authors use robust markers like ERTR5, Keratin 5 that are strongly expressed by mTEC that would clearly separate B cells and mTEC?

The presence of NKT cells in the Rag Ko thymus transplant in Fig 7 demonstrates that these cells have entered the graft from peripheral tissues. This is relevant to the authors comments that thymic PCs are tissue resident. I do not see clear data that shows that experiments involving thymus transplant rule out the possibility that PCs are migrating into the thymus from peripheral tissues.

Overall, the study is of interest, but some of the conclusions that are central to the study are not fully supported by the data.

Reviewer #2 (Remarks to the Author):

In this manuscript, the authors provide convincing evidence that homeostatic serum IgE is largely coming from IgE+ plasma cells located in the thymus in naïve BALB/c mice. However, the fact that these "natural IgEs" promote food allergy by enhancing mast cell survival is in my opinion not demonstrated in this manuscript. The title of the manuscript should therefore be revised, and I have several concerns that should be addressed:

- p2 in the introduction part, the authors should indicate if the findings described have been demonstrated in mice or human. For example, line 2, protective functions for IgE in host defense against venoms or in cancer have been observed in mice, but definitive data in human are still lacking.
- p3 "mice": the background for each mouse strain should be clearly indicated.
- p3 "Flow cytometry": given the fact that the identification of plasma cells is mostly based on FACS data, all antibodies used for flow cytometry should be listed with references and clone numbers, fluorochromes and dilutions used. This could be provided in a table. More details on the specific kits used for IgE, IgA, IgM ELISAs should also be provided.
- p5 : the model described is a model of "food anaphylaxis" rather than "food allergy", as no intestinal features are analyzed (intestinal inflammation, diarrhea..).
- p6 : KN2 mice : for more clarity, the authors should indicate that these mice are knocked-in for human CD2 and that in these mice, cells secreting IL4 express hCD2.
- The use of zsgreen-tbx21 KN2 mice should be discussed in the text of the manuscript, with details on these mice in the methods (data presented in Fig S1C).
- p7 : "to investigate this correlation independently of age, we generated F1 mice between B6 and BALB/c", it is not clear to me why this enables performing correlations independently of age, can you explain ? What is the age of the mice used in Fig 2B ?
- p7 : "while PCs with other isotypes were unaffected (Figure 2E)" : only intracellular IgA is shown in 2E and IL4Ra deficiency seems to double the frequency of IgA+ PCs.
- In figure 1, a significant but only partial reduction in IgE is observed in IL4^{-/-} mice, while chimera experiments in Fig 2 suggest that IL4Ra deficiency completely abrogates IgE expressing cells (2E and 2F). The authors should repeat the experiments shown in Fig 1 using IL4Ra^{-/-} on a BALB/c background, to demonstrate to which extent the response depends on IL4 only or can also rely on IL13 signaling, which is not discussed in the manuscript.
- Figure 2H : indicate exact p values, there is a trend towards reduction in serum IgE in GF AAD mice.
- Data from Fig 3B should be discussed in the text : no IgE+ plasma cells were detected in the bone marrow while IgE+ PCs were detected in the thymus and their numbers increased with age.
- p9 : Icosl^{-/-} mice are not described in the methods.
- In Fig1, the authors show that IgE+ PCs in the thymus are virtually absent in B6 mice. However, it appears that several data in subsequent figures are performed in B6 mice. Therefore it is hard to conclude to which extent these findings could apply to IgE vs other PCs.
- The food anaphylaxis model used by the authors is performed using alum as an adjuvant. Models using alum typically poorly rely on IgE but rather on IgG antibodies and their FcγRs. Therefore, unless the authors perform this model in IgE^{-/-} or FcεRI^{-/-} mice (or at least in mice treated with neutralizing anti-IgE Abs vs isotype control), it is not possible to conclude on the role of IgE and IgE PCs in this model.
- p10. "collectively, these results show that thymus-derived polyclonal IgEs in BALB/c mice expand the number of MCs in the gut at steady state, which consequently promote FA response.". I think this is an overstatement, as the data provided by the authors are only correlations between numbers of IgE/IgE+ PCs and mast cells/anaphylaxis. Again, data on the role of IgE in this food anaphylaxis model are lacking and it is possible that the response (anaphylaxis and mast cell expansion in the gut) is largely IgE-independent, as observed in many food anaphylaxis models.
- It is more clear that PCA responses depend on IgE. However, in Fig 7, the reduced PCA response observed upon transfer of sera from Cd1d^{-/-} mice probably reflects reduced circulating IgE levels in the donor mice, but not the fact that these "natural IgEs expand mast cells" as stated in the title of the figure.
- In the discussion part p11, ref 47 is a completely different model of food allergy using repeated oral challenges to expand mast cell numbers in the gut. This reference can therefore not be used here. "FA response is mainly induced by MC degranulation, which is mediated by FcεRI cross-linking through orally administered antigen". Again, this largely depends on the model used, and very few reports actually show a clear strong requirement for FcεRI in mouse models of food anaphylaxis.
- Fig S2, line 4 of the legend, please replace "left" by "upper panel" and "right" by "lower panel"
- Fig S3. Data in Fig S3B & D are from N=3-4 mice? separate experiments? the sentence "Numbers indicate fold changesNumber indicates frequencies" needs to be corrected.

We thank all two reviewers for their constructive comments, which helped us a lot to improve our manuscript. We have added experiments and revised our manuscript according to the reviewers' comments and suggestions. We highlighted them in yellow and specified questions numbers in the manuscript, which will be deleted in the final version.

Reviewer #1 (Remarks to the Author):

Reports that plasma cells develop IgE that produce IgE. They also show that thymus derived IgEs increase mast cell numbers in peripheral tissues such as gut, which then are responsible for food allergy and anaphylaxis. These IgE producing B cells express Aire, which plays a role in deleting self-reactive CD4 SP thymocytes.

R1Q1) Do the authors have direct evidence that IgE+ B-cells develop intrathymically, or do they migrate into the thymus from peripheral tissues? Can the authors show these cells develop in a 'closed system', such as thymus organ culture?

Response) We thank the reviewer's critical comments. Although we believe that thymic B cells serially mature, it is hard to directly prove that each developmental intermediates migrate from the periphery. As the reviewer suggested, we investigate whether IgE+ B cells develop intrathymically, by FTOC experiment as a closed system. Fetal thymi were cultured with 2'-deoxyguanosine for seven days, and sorted cells were colonized using hanging drop culture and analyzed after seven days. Interestingly, we found that B cells differentiated into PCs in the presence of CD4 T cells. Also, IgE expressing PCs were only detected when NKT2 cells were added. These data are in Supplementary Figures 6 and mentioned in the manuscript (page 8).

R1Q2) A central question not addressed is why is the thymus a specialized place that fosters these IgE producing B-cells? The argument given is that this is connected to the availability of IL4 producing NKT cells in thymus compared to other places. This argument is not strongly supported by the data. IL4+ NKT2 are in fact abundant in other sites, so why the need for a thymic specific population?

Response) As the reviewer pointed out, IL-4 producing NKT2 cells are also present in sites other than the thymus. However, their frequencies are highest in the thymus. In previous studies, we first showed NKT2 cells are the source of homeostatic IL-4¹ and thoroughly investigated the distribution of IL-4 producing NKT2 cells and found they are most abundant in the thymus². Other lymphoid organs, such as the spleen and mesenteric lymph node, IL-4 production from NKT2 cells is much less efficient when we directly compared IL-4 production by human CD2 expression using KN2 mice (right Figure). Hogquist and colleagues suggested that myeloid cells in the thymic medulla present stimulating ligands for NKT2 cells³, but it is unclear why these cells are only present in the thymus. Our experiments that showed IgE producing PCs are

significantly enriched in the thymus is well consistent with the distribution of IL-4 producing NKT2 cells.

R1Q3) Also, Blimp1+CD138+ B-cells are still present at reasonable numbers in NKT-deficient Cd1d KO mice, so the authors need to consider additional intrathymic cellular sources of IL4. Another key experiment here is to add back IL4 into Cd1d KO mice via IV injection – does this restore intrathymic PCs?

Response) We thank the reviewer for raising this critical point. Because the number of IgA and IgG producing PCs was not affected by the absence of iNKT cells as seen in Figure 1C-D, we understand the reviewer's question in underlined part as IgE+ Blimp1+CD138+ B cells. Figure 1C-D showed that both CD1d KO and IL-4 KO mice have similarly decreased (about 20-fold) number of IgE producing PCs in the thymus. Therefore, rather than IL-4, there are additional source of IL-13 that affects the differentiation of IgE-producing PCs. To address this, we first analyzed the total thymocytes of KN2 mice (right Figure), most of IL-4 producing cells in the thymus are iNKT cells, and the majority of CD1d tetramer negative huCD2+ cells are TCR β + CD4 T cells.

Second, we checked the *I4ra*^{-/-} BALB/c mice and found that IgE+ PCs almost completely disappeared, suggesting IL-13 might be the additional signaling affecting the development of IgE-producing thymic PCs. We think the more detailed analysis of IL-13 producing cells in the thymus, other than iNKT cells, seems to be out of the scope of this manuscript. Data from *I4ra*^{-/-} mice are added in Figure 1C-D, and we mentioned this point in our manuscript (page 7)

According to the reviewer's suggestion, we tested the effect of exogenous IL-4 using B6 mice that has a similar frequency of IgE+ cells among total PCs with CD1d KO BALB/c mice. For this, we first tested whether peripheral IL-4s are delivered into the thymus by analyzing phosphorylation of STAT6 (pSTAT6) in splenocytes and thymocytes after 30 minutes of intraperitoneal injection of recombinant mouse IL-4. However, unlike splenocytes, thymocytes failed to upregulate pSTAT6, probably due to the blood-thymus barrier (below Figure A). Also, mouse IL-4 has only about 20 minutes of in vivo half-life. We injected a complex of IL-4 and anti-IL-4 antibodies (11B11), which has dramatically enhanced in vivo efficacy and half-life⁴ to overcome these issues. As shown in the below Figure B (next page), we injected WT B6 mice with IL-4 complex every two days for two weeks, and we analyzed IgE+ plasma cells in the spleen and thymus (Figure C). Eomes expression in CD8 T cells is a hallmark of chronic IL-4 exposure⁵, and we found a robust expression of Eomes, especially in BM CD8 T cells of IL-4 complex injected mice. Accordingly, PCs in BM have increased frequencies of icIgE+ cells. However, in the thymus, Eomes expression was much less efficient, and there were no differences in the development of icIgE+ PCs. Therefore, it is very technically challenging to deliver enough amounts of IL-4 directly into the thymus.

R1Q4) The thymus transplant experiments shown in Fig 3C are confusing. The conclusion is that thymic PCs are tissue resident cell that develop from BM precursors immediately after birth. **R1Q4-1)** The data shown suggests that thymus grafts from 1 week old mice do not contain donor derived thymic PCs. Data is shown for 4 weeks, but what happens if these grafts are harvested earlier? **R1Q4-2)** Surely to support the conclusion drawn, the authors would see thymic PCs in these grafts, which contain BM derived progenitors? **R1Q4-3)** Where is the data in this experiment that shows these cells are tissue resident?

Response) We are sorry for the confusion. The purpose of Figure 3C was to show that thymic PCs are *not* derived from fetal liver progenitors, which are abundant in the newborn thymus. **R1Q4-1)** We transplanted newborn thymi (day 1), which do not have PCs (Figure 3B). It is also possible that B cells derived from fetal liver progenitor would exit the thymus before 4 weeks, as the reviewer suggested to analyze earlier time points. However, we are interested in the B cells in adult thymus and tried to clarify whether they are originated from fetal liver or adult BM. **R1Q4-2)** We found that B cells in the transplanted kidney thymus were mainly derived from CD45.2⁺ host cells (Figure 3C). Because CD45.1⁺ CD19⁺ donor-derived B cells were not detected at all (Figure 3C, middle), we assumed that thymic B cells were mainly derived from BM precursors, but not from fetal liver progenitors. BMT experiments (Figure S7C) further support this conclusion. **R1Q4-3)** The direct evidence that thymic B cells and PCs are tissue resident is from the parabiosis experiments, which showed more than 80-90% of mature B cells and PCs in the thymus were host origin in each parabiont (Figure 3A). We edited page 9 to clarify these points.

R1Q5) The data shown in Fig 6, that thymic PCs express Aire, does not seem to have anything to do with the rest of the study. Do the authors think that their expression of Aire is important for their

development/function, this could be done through analysis of Aire KO mice. Regarding the idea that thymic PCs express Aire, the authors try to demonstrate that these cells belong to the B-cell lineage and not TEC lineage. This is very important because both cell types are thought to express a common range of markers. The microscopy is not particularly clear – can the authors use robust markers like ERTR5, Keratin 5 that are strongly expressed by mTEC that would clearly separate B cells and mTEC?

Response) We thank the reviewer for raising these critical issues. We agree with the reviewer's idea that Figure 6 has no direct relevance with other figures and moved it to Figure S12. In Figure S13, we co-analyzed mTECs, B cells, and PCs at the single-cell level to clarify that thymic PCs are distinct from TECs. We also edited the images for EpCAM to clearly distinguish PCs and mTECs (Figure S12C). Furthermore, we added images using keratin5 (krt5) to separate PCs and mTEC as recommended and confirmed that Blimp1+AIRE+ PCs do not express Keratin5 (Figure S12D). We mentioned these in our manuscript (Page 11).

R1Q6) The presence of NKT cells in the Rag KO thymus transplant in Fig 7 demonstrates that these cells have entered the graft from peripheral tissues. This is relevant to the authors' comments that thymic PCs are tissue resident. I do not see clear data that shows that experiments involving thymus transplant rule out the possibility that PCs are migrating into the thymus from peripheral tissues.

Response) We thank the reviewer for raising this critical issue and are sorry to cause some confusion in data interpretation. Early thymic progenitors (ETPs) from host BM migrate into the graft and reconstruct thymic T cell development upon thymic transplantation. As a result, iNKT cells develop in the thymus from DP thymocytes derived from ETPs, rather than they migrate outside the thymus. Also, as we explained in R1Q4, the parabiosis experiment (Figure 3A) confirms that the majority of thymic PCs are tissue resident. To further investigate whether peripheral PCs can migrate into the thymus, we sorted CD138+CD19+ PCs from spleen, BM, and LNs of congenic CD45.1 mice and transferred them into CD45.2 congenic host and analyzed PCs after seven days. We found that donor-derived cells were detected in the spleen but not in the thymus, indicating that peripheral PCs would be inefficient to migrate into the thymus. These data are presented in Figure S7B and mentioned in our manuscript (page 9).

Overall, the study is of interest, but some of the conclusions that are central to the study are not fully supported by the data.

According to the reviewer's constructive comments and suggestions, we added data and rewrote the manuscript. As a result, we think our manuscript has been significantly improved and hope this resolves most of the reviewer's concerns.

Reviewer #2 (Remarks to the Author):

In this manuscript, the authors provide convincing evidence that homeostatic serum IgE is largely coming from IgE+ plasma cells located in the thymus in naïve BALB/c mice. However, the fact that these "natural IgEs" promote food allergy by enhancing mast cell survival is in my opinion not demonstrated in this manuscript. The title of the manuscript should therefore be revised, and I have several concerns that should be addressed:

We greatly appreciate the reviewer's positive comments and constructive suggestions. We changed our main and sub-title, added additional figures, and edited our manuscript according to the reviewer's comments.

R2Q1) p2 in the introduction part, the authors should indicate if the findings described have been demonstrated in mice or human. For example, line 2, protective functions for IgE in host defense against venoms or in cancer have been observed in mice, but definitive data in human are still lacking.

Response) We thank the reviewer for raising this point. We clarified this point in our manuscript (Page 2).

R2Q2) p3 "mice": the background for each mouse strain should be clearly indicated. p3 "Flow cytometry": given the fact that the identification of plasma cells is mostly based on FACS data, all antibodies used for flow cytometry should be listed with references and clone numbers, fluorochromes and dilutions used. This could be provided in a table. More details on the specific kits used for IgE, IgA, IgM ELISAs should also be provided.

Response) We thank the reviewer for commenting on this critical part. We submitted 'Reporting Summary' that has detailed information of antibodies we used for the analysis. ELISA kits information is also added in the method part of our manuscript.

R2Q4) p5 : the model described is a model of "food anaphylaxis" rather than "food allergy", as no intestinal features are analyzed (intestinal inflammation, diarrhea..).

Response) We are sorry for the confusion of terminology. We agree with the reviewer's comment and changed the term 'allergy' to 'anaphylaxis'.

R2Q5) p6 : KN2 mice : for more clarity, the authors should indicate that these mice are knocked-in for human CD2 and that in these mice, cells secreting IL4 express hCD2.

R2Q6) The use of zsgreen-tbx21 KN2 mice should be discussed in the text of the manuscript, with details on these mice in the methods (data presented in Fig S1C).

Response) As the reviewer pointed out, we clarified the description of KN2 and zsgreen-tbx21 mice in result part of our manuscript (page 7).

R2Q7) p7 : "to investigate this correlation independently of age, we generated F1 mice between B6 and BALB/c", it is not clear to me why this enables performing correlations independently of age, can you explain ? What is the age of the mice used in Fig 2B?

Response) We are sorry for the confusion. In Figure 2A, we used mice aged one to seven weeks old. Because the numbers and frequencies of NKT2 cells increase by age, it is not clear whether the number of IgE+ thymic PCs, serum IgE concentrations, and NKT2 cells are modulated simply by aging. To fix the mice's age at 7-8 weeks old, we performed cross-breeding experiments as seen in Figure 2B-C. We clarified this in the manuscript (page 8).

R2Q8) p7 : "while PCs with other isotypes were unaffected (Figure 2E)" : only intracellular IgA is shown in 2E and IL4Ra deficiency seems to double the frequency of IgA+ PCs.

- In figure 1, a significant but only partial reduction in IgE is observed in IL4^{-/-} mice, while chimera experiments in Fig 2 suggest that IL4Ra deficiency completely abrogates IgE expressing cells (2E and 2F). The authors should repeat the experiments shown in Fig 1 using IL4Ra^{-/-} on a BALB/c background, to demonstrate to which extent the response depends on IL4 only or can also rely on IL13 signaling, which is not discussed in the manuscript.

Response) For more clarity, we added data showing the numbers of IgA+ PCs in Figure 2E and changed the facs plot with a more representative one. In Figure 1B-C, we added data from *Il4ra*^{-/-} BALB/c mice, with very few IgE+ PCs indicating IL-13 signaling also has a role in the differentiation into IgE+ PCs. We pointed this in our manuscript (page 6).

R2Q9) Figure 2H : indicate exact p values, there is a trend towards reduction in serum IgE in GF AAD mice.

Response) As the reviewer pointed out, we rechecked p-values, and there was no significance between the two groups (p=0.066). We indicated each p-value in the Figure 2H.

R2Q10) Data from Fig 3B should be discussed in the text : no IgE+ plasma cells were detected in the bone marrow while IgE+ PCs were detected in the thymus and their numbers increased with age.

Response) We have changed these points in our manuscript according to the reviewer's comments (page 9)

R2Q11) p9 : *Icosl*^{-/-} mice are not described in the methods.

Response) We added a description of this mouse in the method section.

R2Q12) In Fig1, the authors show that IgE+ PCs in the thymus are virtually absent in B6 mice. However, it appears that several data in subsequent figures are performed in B6 mice. Therefore it is hard to conclude to which extent these findings could apply to IgE vs other PCs.

Response) We used *Icosl*^{-/-} and *Cd4*^{cre}*Bcl6*^{fl/fl} mice as B6 mice as we do not have them in BALB/c background. As IL-4 is dispensable for the development of PCs secreting IgA or IgG (Figure 1D), the results from these mice are not applicable for IgE+ PCs. We mentioned this on page 10.

R2Q13) The food anaphylaxis model used by the authors is performed using alum as an adjuvant. Models using alum typically poorly rely on IgE but rather on IgG antibodies and their FcγRs. Therefore, unless the authors perform this model in IgE^{-/-} or FcεRI^{-/-} mice (or at least in mice treated with neutralizing anti-IgE Abs vs isotype control), it is not possible to conclude on the role of IgE and IgE PCs in this model.

- p10. "collectively, these results show that thymus-derived polyclonal IgEs in BALB/c mice expand the number of MCs in the gut at steady state, which consequently promote FA response.". I think this is an overstatement, as the data provided by the authors are only correlations between numbers of IgE/IgE+ PCs and mast cells/anaphylaxis. Again, data on the role of IgE in this food anaphylaxis model are lacking and it is possible that the response (anaphylaxis and mast cell expansion in the gut) is largely IgE-independent, as observed in many food anaphylaxis models.

Response) We thank the reviewer for raising these critical issues. We understand the reviewer's concern and regretfully missed some important aspects of the **food anaphylaxis (FA)** model. Our experimental schematics of the FA model (ova-alum IP twice followed by oral gavage of ova) is from a recent paper ⁶ that showed expanded MCs in the gut promote FA. This paper showed that ILC2s in the gut are expanded by IL-33 derived from injured skin and produce IL-4, consequently expanding **mast cells (MCs)**. Because we showed the correlation between the number of thymic IgE+ PCs, serum IgE concentrations, and the numbers of MCs in the gut, we assumed that thymic IgE increases the number of MCs in the gut via IgE. As the above report showed expanded MCs in the gut promote FA, we also concluded that thymic IgE promotes FA. However, as the reviewer pointed out, we showed some correlations but not definitive evidence demonstrating thymic IgE expanded MCs. For example, we once tried to purify IgEs from thymic PCs and add them back to *Cd1d* KO mice, which was not technically feasible.

We also fully agree with the reviewer's point that FA model we used is not inhibited by blocking IgE. Systemic anaphylaxis in the mouse is mediated by two distinct mechanisms: a classic pathway involving IgE-FcεRI and MCs and an alternative pathway involving IgG-FcγRs and macrophages ⁷. FA model using ova-alum is primarily mediated through the latter, as previously shown ⁸. We also confirmed that IgE blockade in our FA model does not fully reverse FA response, as shown in the below Figure, in which we treated anti-IgE blocking antibody for 3 weeks (A) and measured temperature changes (B)

after oral gavage. Considering these points, we changed our main title, toned down our conclusions, and edited the abstract, introduction, result (page 11-12), and discussion sections (page 13).

R2Q14) It is more clear that PCA responses depend on IgE. However, in Fig 7, the reduced PCA response observed upon transfer of sera from Cd1d^{-/-} mice probably reflects reduced circulating IgE levels in the donor mice, but not the fact that these "natural IgEs expand mast cells" as stated in the title of the Figure.

Response) In PCA response, we transferred sera from WT mice into Cd1d^{-/-} recipient mice. In this setup, Cd1d^{-/-} hosts had a weaker mast cell degranulation as they had fewer mast cells (Figure S14) than WT hosts. We also revised the figure title.

R2Q15) In the discussion part p11, ref 47 is a completely different model of food allergy using repeated oral challenges to expand mast cell numbers in the gut. This reference can therefore not be used here. "FA response is mainly induced by MC degranulation, which is mediated by FcεRI cross-linking through orally administered antigen". Again, this largely depends on the model used, and very few reports actually show a clear strong requirement for FcεRI in mouse models of food anaphylaxis.

Response) We thank the reviewer again for this critical comment. As we explained in R2Q13, we clarified these points and changed references, and rewrote the manuscript.

R2Q15) Fig S2, line 4 of the legend, please replace "left" by "upper panel" and "right" by "lower panel"

Response) We corrected the typos.

R2Q16) Fig S3. Data in Fig S3B & D are from N=3-4 mice? separate experiments? the sentence "Numbers indicate fold changes Number indicates frequencies" needs to be corrected.

Response) We clarified these points and corrected typos.

(References)

- 1 Lee, Y. J., Holzapfel, K. L., Zhu, J. F., Jameson, S. C. & Hogquist, K. A. Steady-state production of IL-4 modulates immunity in mouse strains and is determined by lineage diversity of iNKT cells. *Nature Immunology* **14**, 1146-U1126, doi:10.1038/ni.2731 (2013).
- 2 Lee, Y. J. *et al.* Tissue-Specific Distribution of iNKT Cells Impacts Their Cytokine Response. *Immunity* **43**, 566-578, doi:10.1016/j.immuni.2015.06.025 (2015).
- 3 Wang, H. *et al.* Myeloid cells activate iNKT cells to produce IL-4 in the thymic medulla. *Proc Natl Acad Sci U S A* **116**, 22262-22268, doi:10.1073/pnas.1910412116 (2019).
- 4 Boyman, O., Kovar, M., Rubinstein, M. P., Surh, C. D. & Sprent, J. Selective stimulation of T cell subsets with antibody-cytokine immune complexes. *Science* **311**, 1924-1927, doi:10.1126/science.1122927 (2006).
- 5 Weinreich, M. A., Odumade, O. A., Jameson, S. C. & Hogquist, K. A. T cells expressing the transcription factor PLZF regulate the development of memory-like CD8⁺ T cells. *Nat Immunol* **11**, 709-716, doi:10.1038/ni.1898 (2010).
- 6 Leyva-Castillo, J. M. *et al.* Mechanical Skin Injury Promotes Food Anaphylaxis by Driving Intestinal Mast Cell Expansion. *Immunity* **50**, 1262-+, doi:10.1016/j.immuni.2019.03.023 (2019).
- 7 Finkelman, F. D. Anaphylaxis: lessons from mouse models. *J Allergy Clin Immunol* **120**, 506-515; quiz 516-507, doi:10.1016/j.jaci.2007.07.033 (2007).
- 8 Strait, R. T., Morris, S. C., Yang, M., Qu, X. W. & Finkelman, F. D. Pathways of anaphylaxis in the mouse. *J Allergy Clin Immunol* **109**, 658-668, doi:10.1067/mai.2002.123302 (2002).

REVIEWER COMMENTS

Reviewer #1 (Remarks to the Author):

The authors have substantially revised their manuscript through the inclusion of new data, and the reinterpretation and clarification of key points

Reviewer #2 (Remarks to the Author):

The authors have appropriately addressed my concerns/remarks.

Reviewer #1 (Remarks to the Author):

The authors have substantially revised their manuscript through the inclusion of new data, and the reinterpretation and clarification of key points

Reviewer #2 (Remarks to the Author):

The authors have appropriately addressed my concerns/remarks.

We thank for the reviewers' acceptance of our revised manuscript.